# Uncovering mutation-specific morphogenic phenotypes and paracrine-mediated vessel dysfunction in a biomimetic vascularized mammary duct platform

Matthew L. Kutys [1,2,3,7], William J. Polacheck [1,2,4,7], Michaela K. Welch[1], Keith A. Gagnon [1], Thijs Koorman[5,6], Sudong Kim[1,2], Linqing Li[1,2], Andrea I. McClatchey[5,6] & Christopher S. Chen[1,2]

The mammary gland is a highly vascularized tissue capable of expansion and regression during development and disease. To enable mechanistic insight into the coordinated morphogenic crosstalk between the epithelium and vasculature, we introduce a 3D microfluidic platform that juxtaposes a human mammary duct in proximity to a perfused endothelial vessel. Both compartments recapitulate stable architectural features of native tissue and the ability to undergo distinct forms of branching morphogenesis. Modeling *HER2/ERBB2* amplification or activating *PIK3CA(H1047R)* mutation each produces ductal changes observed in invasive progression, yet with striking morphogenic and behavioral differences. Interestingly, PI3Kα[H1047R] ducts also elicit increased permeability and structural disorganization of the endothelium, and we identify the distinct secretion of IL-6 as the paracrine cause of PI3Kα[H1047R]-associated vascular dysfunction. These results demonstrate the functionality of a model system that facilitates the dissection of 3D morphogenic behaviors and bidirectional signaling between mammary epithelium and endothelium during homeostasis and pathogenesis.

[1] Department of Biomedical Engineering, Boston University, 610 Commonwealth Ave, Boston, MA 02215, USA. [2] Wyss Institute for Biologically Inspired Engineering, Harvard University, 3 Blackfan Circle, Boston, MA 02115, USA. [3] Currently at Department of Cell and Tissue Biology, University of California San Francisco, Box 0512513 Parnassus Avenue, San Francisco, CA 94143, USA. [4] Currently at UNC/NCSU Joint Department of Biomedical Engineering, University of North Carolina, 104 Manning Drive, Chapel Hill, NC 27599, USA. [5] Massachusetts General Hospital Cancer Center, Harvard Medical School, 149 13th Street, Charlestown, MA 02129, USA. [6] Department of Pathology, Massachusetts General Hospital, Harvard Medical School, 149 13th Street, Charlestown, MA 02129, USA. [7] These authors contributed equally: Matthew L. Kutys, William J. Polacheck. ✉email: chencs@bu.edu

Dynamic chemo-mechanical interactions between resident parenchymal cells and the surrounding microenvironment, including stromal cells and the extracellular matrix (ECM), are essential for tissue morphogenesis and homeostasis and are dysregulated in the progression of disease[1–4]. It has been difficult to isolate the effects of these interactions in the complex environment of native tissues in vivo and thus are studied primarily by co-culture of cells on a dish or across a transwell membrane. Recent advances in microphysiological systems engineering and organs-on-chip technology provide methods to miniaturize these interfaces to allow detailed study of interactions between parenchymal and stromal cells and their coordinated responses within controlled environments in vitro[5–7]. However, many of these systems organize cells within static compartments that do not allow cells to freely assemble, remodel, or reciprocally undergo three-dimensional (3D) morphogenesis. Consequently, in vivo models remain the standard for studying tissue morphogenesis and regulatory signaling, despite limitations in the ability to dissect molecular and mechanical mediators of these processes in living animals.

The human mammary gland experiences dramatic morphogenic transitions throughout adult life, capable of undergoing multiple cycles of expansion and regression, with associated changes in cell number, cell composition, and tissue architecture[8,9]. As such, the mammary gland has been established as a multiscale in vivo model to study the mechanisms underlying tissue morphogenesis. During puberty and pregnancy, spatially restricted patterns of cell morphodynamics and proliferation within the mammary epithelium drive ductal extension in response to stromal-derived growth factors, cytokines, and signals from the ECM[9–12]. Reciprocally, the epithelium transmits paracrine signals that direct the fate and behavior of cells in the surrounding microenvironment. In the case of the endothelium lining the local blood vasculature, these paracrine interactions regulate ordered network remodeling, angiogenic expansion, and regression[13]. In breast cancer, increased mutational burden within the developing tumor mass endows mammary epithelial cells with diverse morphogenic behaviors that facilitate invasive transition. These mutations also elicit microenvironmental alterations, in part, through changes in paracrine signaling[14,15]. Investigation of the mechanisms underlying these morphodynamic interactions between epithelial and stromal compartments is essential to the understanding of mammary development, regeneration, and the identification of interventional strategies for the treatment of breast cancer.

A number of 3D mammary culture models have been developed to address the low throughput, relative inaccessibility to experimental manipulation, and visualization difficulties associated with in vivo studies[16]. The development of 3D culture systems using established cell lines to form cystic structures that model the near-spherical ductal terminus or acinus have yielded improved methods for studying normal and tumor-like epithelial architecture and biology[2]. More recently, it has been shown that dissected mouse mammary ducts form polarized cysts or 'organoids' in 3D culture, which can be induced to form multicellular extensions that expand into the ECM and simulate branching morphogenesis during mammary development or oncogenic transformation[11,17–19]. Despite the clear advantage of modeling aspects of the natural architecture of the mammary epithelium, these 3D models pose no control over epithelial architecture and lack a vascular compartment, making it difficult to address these critical aspects of tissue morphogenesis and pathogenesis.

Here, we sought to develop an experimental system capable of recapitulating and dissecting diverse 3D mammary morphogenic processes and complex paracrine interactions amongst multiple cell types. To enable this longer-term vision, in this study we focus first on building and studying the interactions between two principle tissue structures of the mammary gland—a biomimetic human mammary epithelial duct cultured in proximity to an endothelialized vessel. These compartments are designed to stably mimic aspects of in vivo architecture, including a growth arrested ductal epithelium lining a fluid-filled luminal compartment and a perfusable endothelium within 3D ECM, as well as to recapitulate diverse modes of tissue morphogenesis in each compartment. We employ this platform to deconstruct the role of specific mutations in driving important invasive, tissue morphodynamic changes that together highlight a biomimetic system to dissect the molecular and physical principles underlying the crosstalk between vascular and mammary epithelial compartments during 3D tissue remodeling.

## Results

**A vascularized mammary duct capable of tissue morphogenesis.** To study the morphodynamics of a 3D mammary ductal epithelium and its reciprocal communication with vasculature, we engineered a microfluidic platform in which a perfused endothelium-lined channel passes in close proximity to a lumenized mammary epithelial duct, both surrounded by 3D ECM. The platform was assembled by polymerizing a type-I collagen hydrogel of physiologically relevant stiffness[20] into a chamber inside a poly(dimethylsiloxane) (PDMS) mold in the presence of two needle templates (160-μm diameter), one passing through the chamber to form a perfused, hollow channel and the second inserted into the chamber to form a dead-ended channel positioned perpendicularly 500 μm away, referred to subsequently as 'vascular' and 'duct' channels, respectively (Supplementary Fig. 1a). Injection of the human mammary epithelial cell line MCF10A into the duct channel failed to promote stable ducts as cells invaded into the interstitial collagen type I matrix (Supplementary Fig. 1b). We therefore examined whether introduction of basement membrane ECM could induce cells to form a stable duct by injecting basement membrane ECMs into the duct channel to passively coat its surface prior to introduction of cells. Purified basement membrane proteins such as laminin or collagen IV alone did not promote the assembly of noninvasive ducts, but dilute growth factor reduced Matrigel (GFR MG)-coated devices formed a stable, hollow duct (Supplementary Fig. 1b). Similarly, injection of primary human dermal microvascular cells (hMVECs) into an uncoated central collagen channel formed a confluent functional vascular endothelium[21,22] (Fig. 1a, b, Supplementary Fig. 1a). The vasculature was maintained under perfusion for the duration of co-culture and, in the absence of environmental or genetic perturbation, these two tissues remained stable for up to two weeks (Supplementary Fig. 1c).

In traditional 3D acinar MCF10A models, cells proliferate to form a multicellular aggregate wherein cells in the central space undergo apoptosis to give rise to hollow acini. In the system here, seeded cells quickly populate the GFR MG/collagen surface of the ductal channel and excess cells sterically blocked from making ECM contact underwent progressive clearance by apoptosis (Supplementary Fig. 2a). While MCF10A are limited in their ability to form mature apical domains[23], cells in the duct established apical–basal polarity over a seven day time course as evidenced by the reorganization of the Golgi marker GM130 towards the luminal face (Supplementary Fig. 2b, Fig. 1b, c), the basal localization of the basement membrane receptor α6 integrin, the basolateral localization of E-cadherin (Fig. 1b), and columnar cellular restructuring (Supplementary Fig. 2c). The resulting stable, noninvasive monolayers within the channels displayed growth arrest as evidenced by minimal actively proliferating Ki67+ cells (Fig. 1b), a process that is dependent upon stable cell–cell contacts in MCF10A[24–26]. Thus, despite the limited maturation capabilities of MCF10As, this

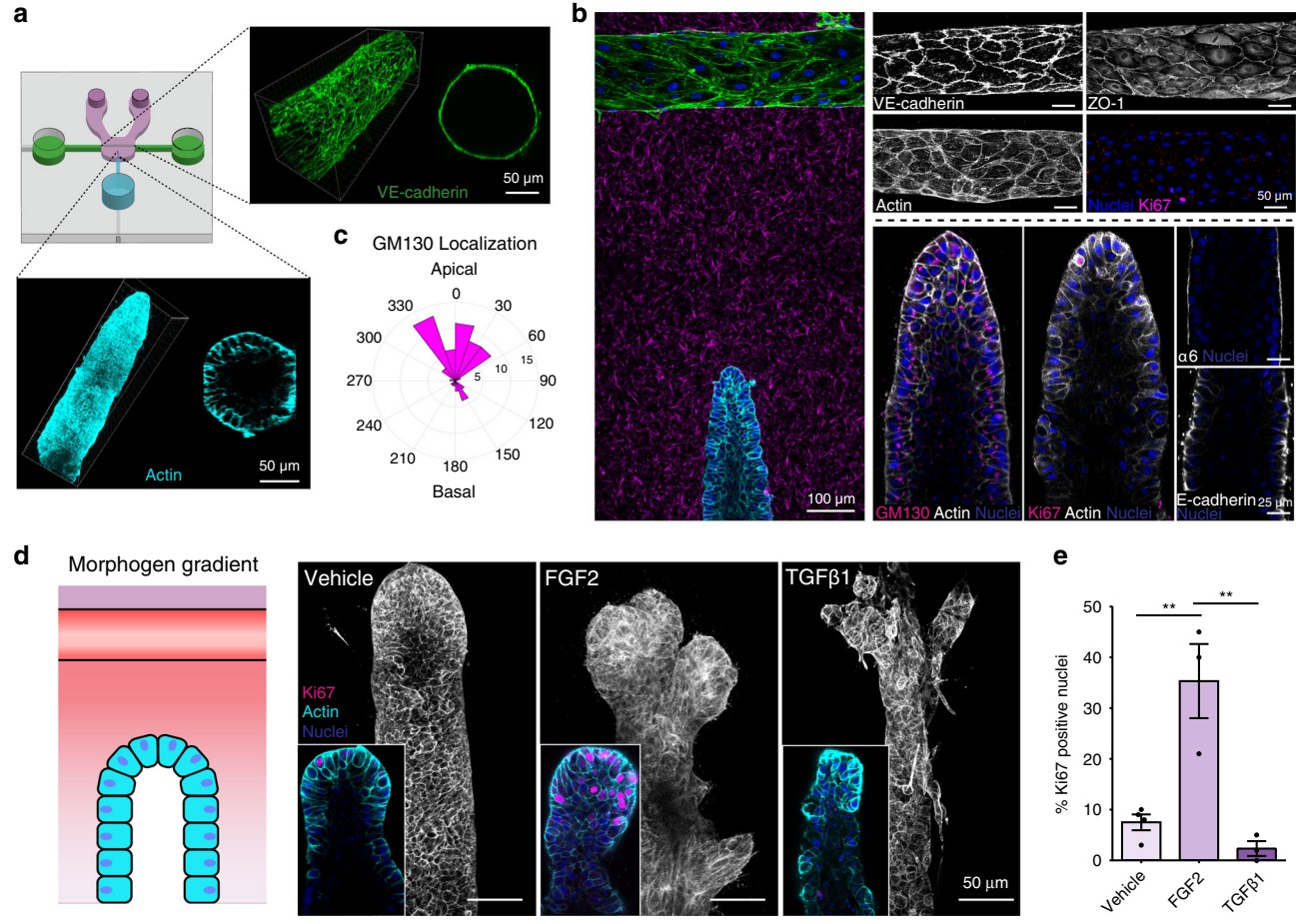

**Fig. 1 A vascularized mammary duct platform models native 3D tissue architectures and morphogenesis.** Tissues are at one week of co-culture unless otherwise indicated. **a** Organotypic microfluidic device consisting of an engineered 3D endothelial vessel (green) and mammary epithelial duct (cyan) in physiologic ECM (magenta), enabling long-term culture and paracrine signaling. Inset: 3D reconstruction of endothelial vessel (green—VE-cadherin) and epithelial duct (cyan—phalloidin). **b** Left: Composite stitched maximum intensity projection micrograph of tissue base to medial section of a mammary epithelial duct (cyan, phalloidin) and proximal vasculature (green, phalloidin); Alexa Fluor-labeled collagen I (magenta), DAPI (blue). Top right: Representative maximum intensity micrographs of mature endothelial vessels immunostained for VE-cadherin, ZO-1, actin, and Ki67, as indicated. Bottom right: Individual confocal slice micrographs from 3D epithelial duct midsections immunostained for GM130, Ki67, α6 integrin, E-cadherin, actin, and DAPI, as is indicated in each panel. **c** Quantification of golgi localization as measured by the nuclear-GM130 axis ($n = 52$ cells examined from three independent ducts). 0 corresponds to a nuclear-GM130 axis apically oriented toward the lumen, while 180 corresponds to a basal orientation. **d** Cartoon depicting morphogen gradients established by perfusing growth factors through an acellular vascular channel. Immunographs of quiescent MCF10A ducts exposed to vehicle (DMSO), FGF2 (3 nM), or TGFβ1 (5 ng/ml) gradients for three days. Inset: high magnification slice micrograph of duct terminus immunostained with phalloidin (cyan), DAPI (blue) and Ki67 (magenta). **e** Quantification of percentage of Ki67 positive nuclei in ducts treated with each morphogen gradient ($n = 4, 3, 3$ ducts examined across three independent experiments; one-way ANOVA with Bonferroni post test Vehicle vs FGF2 **$p = 0.0036$, FGF2 vs TGFβ1 **$p = 0.002$, mean ± s.e.m). All images are representative of at least three independent experiments. Source Data are provided as a Source Data file.

approach resulted in an epithelial cell-lined dead-ended channel, reminiscent of an anatomical ductal lumen. After one week of co-culture, cells in the endothelialized channel exhibited tight vascular barrier and cell–cell adhesion, as demonstrated by measuring the extravasation of fluorescently labeled 70 kDa dextran[21,22] (Supplementary Movie 1) and the junctional localization of VE-cadherin, ZO-1, and F-actin (Fig. 1b), and showed minimal active proliferation (Fig. 1b). Stable development of the endothelial[21,22,27] and epithelial (Fig. 1d, Supplementary Fig. 2d) tissue compartments was not dependent on, or detrimentally affected by, co-culture. Thus, this platform appeared to provide a means to recapitulate the basic architectures of a rudimentary mammary duct and microvascular endothelium in a physiologically relevant, closed co-culture system.

Having established quiescent ductal tissues, we next tested their morphogenic capacity by introducing defined microenvironmental

gradients of morphogens between the two compartments (Supplementary Fig. 3) to stimulate normal and pathologic morphodynamic transitions in our model. MCF10A cells were seeded into duct channels and cultured until a growth arrested confluent monolayer formed, at which point linear growth factor gradients were introduced for three days by diffusion from an acellular vascular channel under perfusion (Fig. 1d, Supplementary Fig. 2e). Exposure to a gradient of FGF2 led to striking expansion of the duct terminus toward the gradient source. This expansion was driven by cell proliferation and resulted in the filling of the lumen, broad ductal expansion, and cell multilayering restricted to the ductal ends (Figs. 1d, 1e, Supplementary Fig. 2f), similar to what has been described during in vivo mammary tissue elaboration and branching in FGF2-treated mouse mammary organoids[11,28]. Conversely, exposure to a gradient of TGFβ1 led to severe ductal constriction, preservation of the lumen, and an invasive phenotype

suggesting epithelial-to-mesenchymal transition at the ductal end, directed toward the gradient source, that was independent of proliferative activity (Fig. 1d, e, Supplementary Fig. 2e, f). Together, these studies demonstrate a platform that supports the growth and function of a quiescent, vascularized mammary duct and permits high-resolution monitoring of emergent cellular behaviors during diverse tissue morphogenic events.

**Vascular remodeling via mammary duct paracrine signaling.** Along with the cyclical growth and differentiation of the mammary epithelium, the vasculature of the mammary gland undergoes repeated cycles of expansion and regression which are partially dependent on paracrine secretion of cytokines, hormones, and growth factors from the epithelium[29]. Having established the ability to stably co-culture a quiescent vasculature and mammary duct in our platform, we investigated whether alterations in the mammary epithelium disrupt this quiescence and elicit changes in the nearby endothelium. Vascular endothelial growth factor (VEGF)-A (herein referred to as VEGF) is a potent angiogenic factor expressed during pregnancy[30] and frequently expressed during breast cancer transformation[31]. To examine whether our model could be used to study such interactions, we generated ducts composed entirely of cells expressing VEGF-IRES-GFP or control IRES-GFP (Fig. 2a, Supplementary Fig. 4a). Phase-contrast

and confocal microscopy were used to assess endothelial remodeling over three days in response to paracrine factor gradients generated from the epithelium (Supplementary Fig. 3). In comparison to GFP controls, VEGF expressing cells induced significant vascular remodeling and vessel dilation (Fig. 2a, b, Supplementary Fig. 4b), with no obvious effect on mammary duct structure or behavior. Over time, this phenotype progressed further to the formation of filopodia-like protrusions and nascent angiogenic sprouts (Fig. 2c, d, Supplementary Fig. 4b). Examination of these effects at high magnification showed that VEGF secretion from the duct triggered endothelial cell elongation and a marked shift in actin organization from a cortical, junctional localization to transverse stress fibers spanning the cell body (Fig. 2d), indicative of destabilized endothelial cell junctions[22,32,33]. Treatment of vessels with conditioned medium from VEGF cells similarly elicited vascular morphogenesis compared to GFP control (Supplementary Fig. 4c). Importantly, treatment of vessels with basal mammary duct medium (BM) alone did not result in the vascular phenotypic changes (Fig. 2c) that were observed with the conditioned media experiments (Supplementary Fig. 4c). Together, these data prescribe this phenotype to VEGF paracrine signaling. We next tested whether delivery of a targeted inhibitory agent to the vasculature would ameliorate the endothelial response to VEGF produced by the mammary epithelium. The VEGF receptor 2 (VEGFR2) inhibitor,

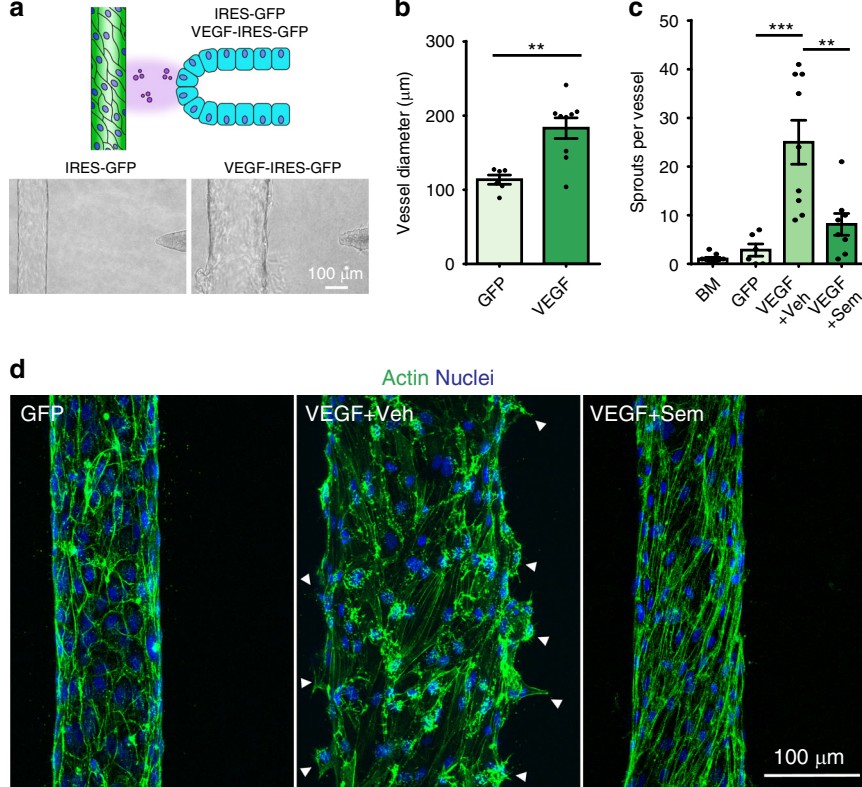

**Fig. 2 Vascular morphogenesis driven by mammary paracrine signaling. a** Cartoon of co-culture paracrine experimental setup. MCF10A cells expressing IRES-GFP or VEGF-IRES-GFP were seeded into the duct channels adjacent to an endothelialized vessel. Phase-contrast images of vessels with control (left) and VEGF expressing ducts (right) after two days in co-culture. **b** Quantification of endothelial vessel diameter after three days of culture with either GFP or VEGF expressing MCF10A cells ($n = 6, 9$ vessels examined across three independent experiments; two-tailed, unpaired Student's $t$ test **$p = 0.0019$). **c** Quantification of sprout number per endothelial vessel after three days of culture with acellular basal assay medium (BM), GFP expressing ducts (GFP), or VEGF expressing ducts in which DMSO (VEGF + Veh) or 10 μM Semaxanib (VEGF + Sem) was delivered to the vasculature. ($n = 9, 6, 9, 8$ vessels examined across three independent experiments; one-way ANOVA with Bonferroni post test ***$P = 0.000014$, **$P = 0.0011$). **d** Maximum intensity projection micrographs of tissue base to medial section of endothelial vessels co-cultured with GFP or VEGF ducts and treated with DMSO or 10 μM Semaxanib; phalloidin (green) and DAPI (blue). Arrows indicate nascent angiogenic sprouts. For all plots, values mean ± s.e.m and all images are representative of at least three independent experiments. Source Data are provided as a Source Data file.

Semaxanib[34], was perfused within the endothelial vessel overnight between days two and three of the time course. Consistent with previous observations, Semaxanib arrested the formation of VEGF-driven cellular protrusions that are critical to the initiation and growth of angiogenic sprouts[27]. Despite a reversion of vascular dilation (Fig. 2d, Supplementary Fig. 4d), Semaxanib did not fully reverse VEGF-driven changes in endothelial cell morphology and actin organization (Figs. 2c, d), possibly because prolonged VEGF exposure produced cumulative endothelial restructuring that is not immediately reversible following inhibition of VEGFR2 signaling. This experimental platform therefore reconstructs the paracrine crosstalk between epithelial and endothelial compartments critical to mammary tissue biology and permits detailed description of the resulting cell and tissue morphogenic behaviors.

**Distinct morphogenic consequences of breast cancer mutations.** Genome-wide sequencing efforts have identified many recurring breast cancer mutations, however it remains largely unclear how each mutation leads to the specific cellular behaviors that underlie tumor invasion and metastasis[35,36]. The ErbB-PI3K-Akt signaling axis is the most prominently altered pathway in human breast cancers[37], yet despite the established hierarchical signaling relationship between ErbBs, PI3K, and Akt, it remains poorly understood how these mutations confer mammary epithelia with invasive potential[35,36] or how they might elicit changes in the surrounding tissue microenvironment, including tortuous, angiogenic, and highly permeable tumor vasculature[38,39]. To begin to investigate whether specific mutations within this axis contribute to distinct epithelial morphogenic changes and nonautonomous vascular effects, we modeled two of the most common patient-derived breast cancer somatic genetic alterations from this pathway in isogenic, nontumorigenic MCF10A cells: ErbB2 receptor amplification (ErbB2$^{amp}$, by wild-type *HER2/ERBB2* over-expression) or a PI3Kα mutation (PI3Kα$^{H1047R}$, by *PIK3CA (H1047R)* expression) that renders PI3Kα constitutively active (Fig. 3a, b). Mutant cells were mixed with wild-type cells at a 1:10 ratio before seeding to generate mosaic ducts, permitting the emergence and observation of aberrant mutant behaviors within the architecture of an otherwise normal duct. Although both mutants exhibited an invasive phenotype in comparison to transduction control empty vector ducts (EV), including filling of the ductal lumen and disordered invasive migration into the ECM, we observed key morphological and behavioral differences in rates and modality (Fig. 3b, Supplementary Fig. 5a–f). Analyzing the ducts after one week, invasion of PI3Kα$^{H1047R}$ ducts occurred more rapidly than ErbB2$^{amp}$ (Supplementary Fig. 5d) and had distinct mesenchymal morphology, commonly invading as single cells with lamellipodia-driven protrusions, as well as mesenchymal transcriptional and adhesive signatures (Supplementary Fig. 5g, h). ErbB2$^{amp}$-driven invasion was collective ameboid, characterized by a bulbous, invasive cellular cohort (Fig. 3b, Supplementary Fig. 5a–e). These morphogenic differences manifested in the presence or absence of nearby co-cultured vasculature (Supplementary Fig. 5i, j). Interestingly, immunostaining for mutant cells via the associated hemagglutinin (HA)-tag revealed that while both mutants drive invasion, the cellular composition of the invasive fronts were distinct. ErbB2$^{amp}$ cohorts were predominantly mutant cells and PI3Kα$^{H1047R}$ fronts contained nearly equal numbers of wild-type cells (Supplementary Fig. 5j, k), suggesting nonautonomous effects of the PI3Kα$^{H1047R}$ mutation. These results are consistent with clinical observations that deregulation of ErbB-PI3K signaling elements can elicit distinct responses[40] and highlights the ability of this platform to reveal specific morphogenic behaviors associated with distinct breast cancer-associated genetic alterations.

Having observed distinct invasive behaviors between ErbB2$^{amp}$ and PI3Kα$^{H1047R}$ ducts, we next investigated their effects on the adjacent vasculature. After one week, independent of direct cell interaction, co-culture with either mutant duct led to moderate vascular dilation in comparison to EV ducts (Fig. 3b, d). However, when we analyzed vascular barrier function in each ductal microenvironment, we observed that PI3Kα$^{H1047R}$ ducts led to a significant increase in permeability in comparison to EV and ErbB2$^{amp}$ ducts (Fig. 3c, Supplementary Video 1). Close inspection of PI3Kα$^{H1047R}$-associated vessel architectures revealed a dramatic shift in F-actin localization from tightly associating with cell–cell junctions to diffuse, cytoplasmic localization (Fig. 3b, e), along with a weakening of VE-cadherin-containing adherens junctions (Supplementary Fig. 6a, b), changes in two critical mediators of vascular barrier supporting the observed increase in vessel permeability[22]. Therefore, it appears that ErbB2$^{amp}$ and PI3Kα$^{H1047R}$ may not only impact mammary epithelia via distinct behavioral effects, but also by differentially driving vascular dysfunction.

**PI3Kα$^{H1047R}$ leads to IL-6 secretion and vascular dysfunction.** We identified dysregulated endothelial architecture and increased permeability by PI3Kα$^{H1047R}$ ducts notably in the absence of any direct interaction with the invading epithelial cells. Therefore, we next investigated if specific paracrine factors from PI3Kα$^{H1047R}$ ducts were responsible for driving this vascular phenotype. After finding that this phenotype was independent of VEGF secretion (Supplementary Fig. 7c), we surveyed the secreted cytokine profiles of conditioned media isolated from EV, ErbB2$^{amp}$, and PI3Kα$^{H1047R}$ cells. While several cytokines were similarly upregulated in the two mutants compared to EV control, only increased interleukin-6 (IL-6) secretion was unique to PI3Kα$^{H1047R}$ (Supplementary Fig. 7a, b, d). Indeed, western blots of PI3Kα$^{H1047R}$ conditioned medium confirmed an approximate threefold increase in secreted IL-6 protein (Fig. 4a, Supplementary Fig. 7d).

To examine whether the PI3Kα$^{H1047R}$-associated vascular defects were due to the secretion of IL-6, conditioned media from EV or PI3Kα$^{H1047R}$ ducts were collected, concentrated, and added to stabilized endothelial vessels from an acellular ductal compartment overnight (Fig. 4b). Overnight treatment alone with PI3Kα$^{H1047R}$ conditioned medium led to increased vascular permeability, decreased cortical actin, and angiogenic response in comparison to EV conditioned medium (Fig. 4b–d). We then explored whether addition of recombinant human IL-6 (rhIL-6) alone was sufficient to drive these responses. Surprisingly, treatment of vessels with basal duct culture medium containing rhIL-6 (BM + rhIL-6) resulted in no obvious differences in vascular permeability or actin organization (Fig. 4c, d). However, supplementing EV conditioned medium with rhIL-6 triggered defects in vessel permeability, actin organization, and sprouting behavior that was not observed with EV conditioned medium alone (Fig. 4b–d). For signaling activity in most contexts, IL-6 requires the secreted form of human interleukin 6 receptor subunit α (IL-6Rα)[41], which we found to be uniformly secreted in all duct backgrounds but absent from the basal duct culture medium (Fig. 4a, Supplementary Fig. 7d). Together, these data suggest a putative mechanism linking somatic PI3Kα$^{H1047R}$ mutation in mammary epithelia to elevated IL-6 secretion that in turn drives endothelial dysregulation.

**Discussion**

Our understanding of the biochemical and mechanical interactions between parenchymal and vascular tissues that shape and

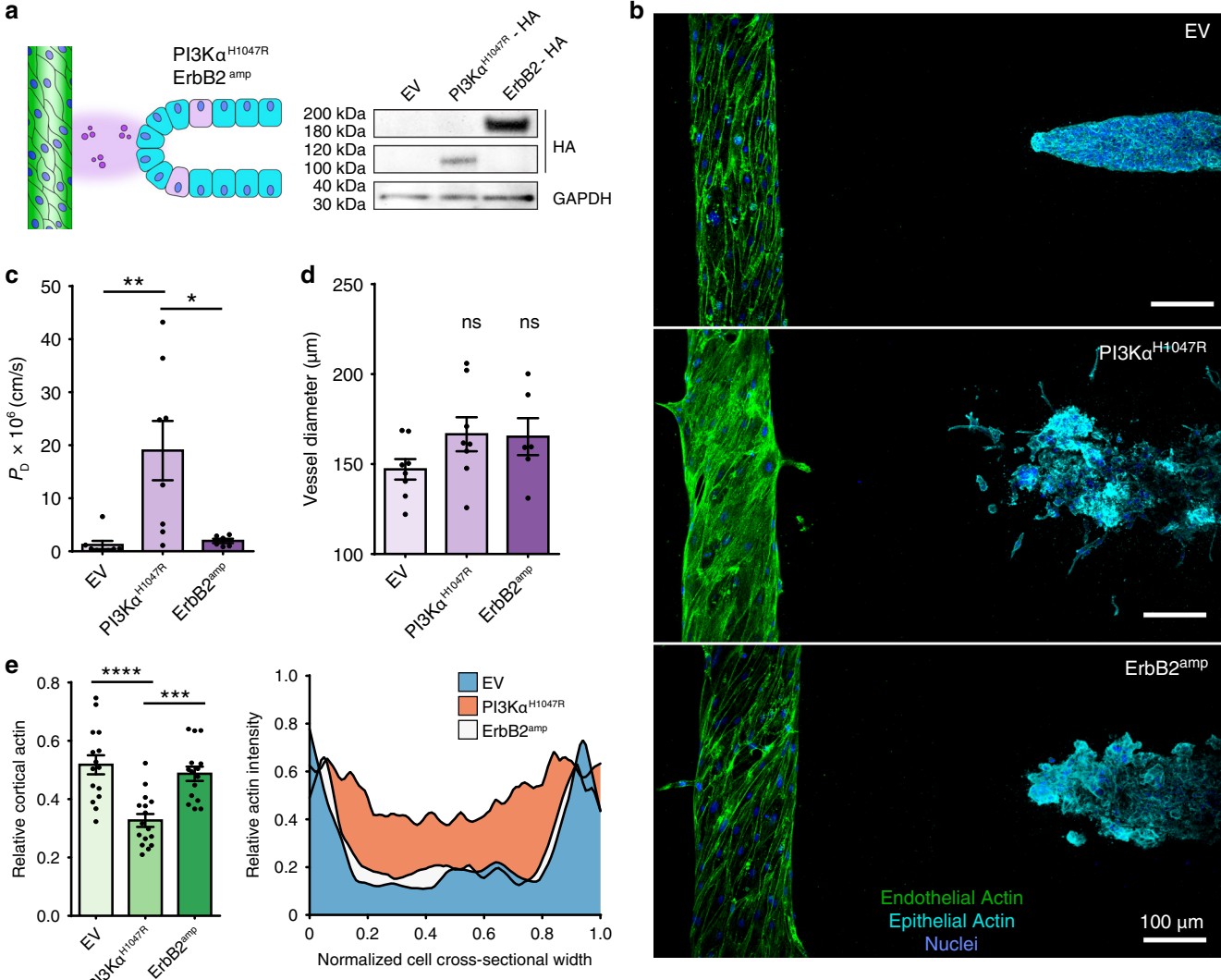

**Fig. 3 Ductal and vascular consequences resulting from specific breast cancer mutations. a** Left: Cartoon of co-culture paracrine experimental setup with mosaic mutant/wild-type ducts. Ducts were generated from MCF10A cells stably expressing empty vector (EV), wild-type hemagglutinin (HA)-tagged ErbB2 (ErbB2amp), or constitutively active HA-tagged PI3Kα H1047R (PI3KαH1047R) along with each associated vascular compartment. Right: Western blot of lysates from EV, ErbB2amp, or PI3KαH1047R MCF10A. **b** Composite stitched maximum intensity projection micrographs of endothelial vessels co-cultured for five days along with their corresponding modified ducts. Endothelial vessels: phalloidin (green), DAPI (blue), epithelial ducts: phalloidin (cyan), DAPI (blue). **c** Diffusive permeability ($P_D$) of 70 kDa dextran across the endothelial barrier ($n = 8, 8, 6$ vessels examined across three independent experiments; one-way ANOVA with Bonferroni post test, ** $P = 0.0049$, * $P = 0.012$) and (**d**) endothelial vessel diameter in each co-culture setting ($n = 8, 8, 6$ vessels examined across three independent experiments; one-way ANOVA with Bonferroni post test). **e** Left: Quantification of cortical actin measured from phalloidin labeled micrographs ($n = 12, 13, 13$ cells taken from three independent vessel experiments one-way ANOVA with Bonferroni post test, **** $P = 7.6845e{-}06$, *** $P = 0.0003$). Right: Averaged, cross-sectional actin distribution in endothelial vessels co-cultured with specific mutant ducts ($n = 11$ cell actin profiles taken from three independent vessel experiments). For all plots, values mean ± s.e.m. and all images are representative of at least three independent experiments. Source Data are provided as a Source Data file.

contribute to organ function has depended largely on in vivo model systems, where identifying causality and the contributions of specific molecules and cells can be challenging. In vitro platforms developed to study these interactions in the mammary gland have cultured mammary acini with disaggregated stromal endothelial cells, or have presented the two populations via compartmentalized, 2D substrates[26,42–44]. Microphysiological systems have been engineered to analyze these interactions in the context of mammary tumor biology by embedding tumor aggregates or single cells in 3D ECM containing a microvascular network, and these systems have been applied largely in tumor viability, angiogenesis, and drug sensitivity assays[45–48]. The platform presented here advances upon these models by not only

mimicking native architectures and physiologic tissue–tissue interfaces necessary for paracrine crosstalk, but also by faithfully recapitulating diverse tissue morphogenic processes and paracrine signaling in response to specific environmental and genetic stimuli, enabling high-resolution monitoring and mechanistic dissection of emergent behaviors.

A distinctive feature of our platform is the ability to capture different stages of mammary epithelial morphogenesis and invasive progression, including initial changes in mammary cell and tissue organization, progressive tissue branching or invasion, and coordinated changes in vascular function and remodeling. The model faithfully captured qualitatively distinct morphogenic phenotypes in response to specific soluble growth factors such as

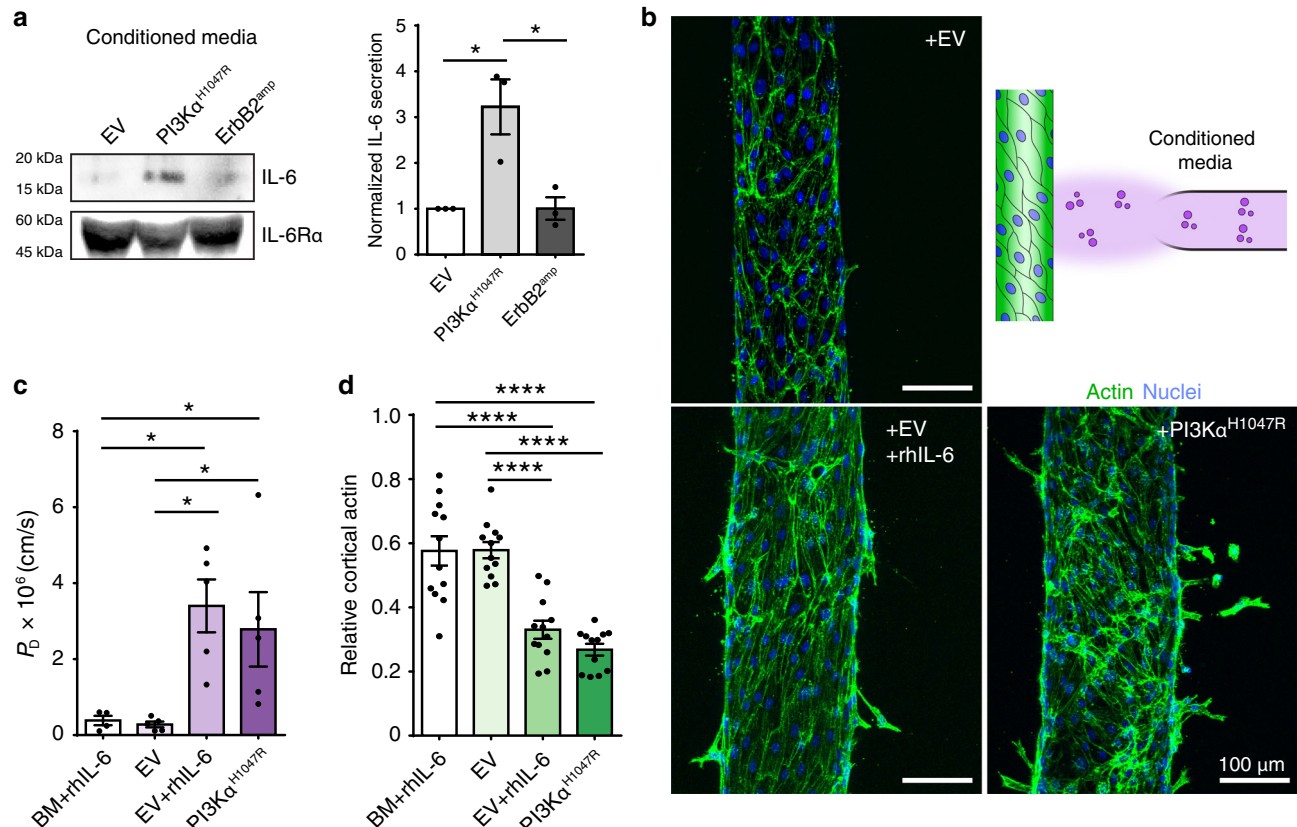

**Fig. 4 PI3Kα^H1047R increases IL-6 secretion to drive vascular dysfunction and remodeling. a** Representative western blot and associated quantification of conditioned media from EV, ErbB2^amp, and PI3Kα^H1047R ducts immunoblotted for human IL-6 and IL-6Rα ($n = 3$ western blots from three independent experiments). **b** Top right: Experimental schematic of endothelial vessel-conditioned media setup. Maximum intensity projection micrographs of endothelial vessels treated overnight with EV, EV+ recombinant human IL-6 (rhIL-6) (200 ng/ml), or PI3Kα^H1047R conditioned media; phalloidin (green), DAPI (blue). **c** Diffusive permeability P_D ($n = 4, 5, 5, 5$ vessels examined across three independent experiments; one-way ANOVA with Bonferroni post test, BM + rhIL-6 vs EV + rhIL-6 *$P = 0.019$, BM + rhIL-6 vs PI3Kα^H1047R *$P = 0.035$, EV vs EV + rhIL-6 *$P = 0.0104$, EV vs PI3Kα^H1047R *$P = 0.0205$) and (**d**) relative cortical actin measured from vessels treated overnight with basal assay medium supplemented with 200 ng/ml rhIL-6 (BM + rhIL-6) or EV, EV + rhIL-6, or PI3Kα^H1047R conditioned media ($n = 12, 12, 12, 12$ cells examined from three vessel experiments; one-way ANOVA with Bonferroni post test, BM + rhIL-6 vs EV + rhIL-6 ***$P = 1.32402e-06$, BM+rhIL-6 vs PI3Kα^H1047R ****$P = 1.08309e-08$, EV vs EV + rhIL-6 ****$P = 1.0819e-06$, EV vs PI3Kα^H1047R ****$P = 8.84689e-09$). For all plots, values mean ± s.e.m. and all images are representative of at least three independent experiments. Source Data are provided as a Source Data file.

FGF2 versus TGFβ1 (Fig. 1d). While incomplete apical maturation that is characteristic of MCF10A cells may potentiate the responses observed, the distinct morphogenic responses to FGF2 and TGFβ1 are consistent with those described in vivo and in other systems[11,19,28], indicating the potential utility in the model to understanding intrinsic mechanisms underlying mammary morphogenic diversity.

Similarly, how specific breast cancer-associated genetic alterations impact the morphogenic and invasive behaviors of mammary epithelial cells in a native tissue setting remains largely unknown. Here, our model revealed previously unappreciated morphologic effects of ErbB2^amp versus PI3Kα^H1047R ducts. ErbB2^amp ducts invaded more slowly, consistent with recent studies that suggest ErbB2-amplified ductal carcinoma in situ may take longer to progress despite ultimately more aggressive behavior[49]. ErbB2^amp invasive fronts displayed behaviors characteristic of collective ameboid migration that is classically associated with reduced cell-matrix adhesion (Fig. 3b, Supplementary Fig. 5a–e)[50]. In line with these findings, ErbB2 overexpression causes anoikis resistance that contributes to luminal filling in mammary acini[51]. Interestingly, ErbB2-mediated anoikis suppression is reported to require the ErbB2 sequestration of, and activity of, α5 integrin[52] and may point toward a complex

interplay between ErbB receptors and integrin signaling governing this 3D invasive mode. Conversely, PI3Kα^H1047R ducts invaded more rapidly, adopting a combination of elongated single and collective mesenchymal behaviors (Fig. 3b, Supplementary Fig. 5a–e) with reduced E-cadherin expression (Supplementary Fig. 5g, h), suggesting a greater dependence on cell-matrix adhesion. While PI3Kα activating mutations increase mesenchymal gene profiles in mammary epithelia, their enhanced invasive potential is reportedly independent of downstream Akt/mTOR signaling pathways[53]. Further, recent studies have linked activating PI3Kα mutations to enhanced Rac1 and Yap signaling during breast cancer progression in vivo[54,55] and suggest an invasive program that may be potentiated by mechanical alterations to the ECM that commonly occur during breast tumor progression. Overall, the qualitatively distinct invasive phenotypes of each mutant suggest the important possibility that there are multiple paths to metastatic progression, and that these paths may be mechanistically and therefore clinically and therapeutically distinct.

Importantly, interactions between different cellular compartments play a major role in tissue morphogenesis and disease, and the studies here provide an approach to begin to study such interactions. Structural abnormalities and increased permeability

are hallmarks of tumor-associated vasculature that can feedback to influence cancer growth, predisposition to metastasis, and drug delivery[56]. The excessive leakiness of tumor vessels permits dysregulated infiltration of immune cells and activated platelets into the tumor microenvironment, imparts fluid stresses on tumors, and causes a rise in interstitial fluid pressure that limits drug transport[57]. These factors contribute to the inflamed tumor microenvironment, induce tumor invasive responses and, along with disruption of barrier function and surface presentation of adhesive ligands in the vasculature, facilitate metastatic dissemination upon vascular interaction[57]. Distinct molecular subtypes of breast cancer have been differently correlated with signatures of vascular dysregulation[58,59], highlighting a need to isolate the precise underlying genetic alterations and associated molecular signals that drive these vascular changes. Intriguingly, in concert with distinct morphogenic behaviors, we observed that PI3Kα$^{H1047R}$ ducts specifically triggered increased permeability in the adjacent vasculature through the secretion of IL-6 (Fig. 4), which has previously been linked to autonomous STAT3 activation resulting from oncogenic *PIK3CA(E545K)* and *(H1047R)* mutations in mammary epithelia[60]. In addition to the vasculature, this putative connection between IL-6 and PI3Kα$^{H1047R}$ raises implications for paracrine crosstalk with resident immune, adipocyte, and mesenchymal cells that advances breast cancer progression in patients harboring PI3Kα activating mutations[61,62]. Overexpression of ErbB2 in human tumors has been associated with increased angiogenesis and VEGF expression[63], yet neither were observed in our system. One potential explanation is that the notable secretory changes that we observed in ErbB2$^{amp}$ ducts require additional stromal components, such as fibroblasts or resident macrophages, to trigger vascular morphogenic cascades. Thus, these studies illustrate the power of our platform to begin to deconstruct how specific breast cancer-associated genetic alterations contribute to key changes in vascular function.

The organotypic model system described here was designed to meet the challenge of engineering progressively more complex tissue models while still remaining grounded in mechanistic approaches and capturing important features of a native vascularized mammary duct. Here, we illustrate its utility by employing it to discover new insights into the cellular processes that drive mammary epithelial morphogenesis, how specific oncogenic changes influence these processes, and importantly how signaling between ductal and vascular endothelial compartments can further drive tissue-level morphogenic changes. The incorporation of tunable biomaterials where ECM biophysical parameters can be precisely controlled will allow study of the influence of ECM mechanics on 3D tissue architecture and polarity, complex morphogenic behaviors, and paracrine responses. Further, the introduction of additional parenchymal cell types will extend the platform's utility to other developmental settings and cancers, while the incorporation of components of whole blood and other stromal cells, mural cells, and immune cells could deepen our understanding of the complex bidirectional paracrine cascades behind breast tumor progression, mammary inflammation, and other pathologies. Ultimately the power of such organotypic platforms lies in the ability to construct physiologically relevant compartments and architectures, while also supporting the remodeling that is central to tissue development and disease. This study adds to the growing array of engineered biomimetic platforms that facilitate the reconstitution and mechanistic dissection of multicellular, heterotypic 3D morphogenic events that were traditionally only accessible in in vivo models.

## Methods

**Cell culture**. Human mammary epithelial line MCF10A (ATCC) were cultured in growth medium, consisting of DMEM/F12 (1:1, Gibco) supplemented with 5%

horse serum (Invitrogen), 20 ng/ml rhEGF (Peprotech), 0.5 mg/ml hydrocortisone (Sigma), 100 ng/ml cholera toxin (Sigma), 10 μg/ml insulin (Sigma) and 1% penicillin/streptomycin (Life Technologies)[26]. For co-culture experiments, MCF10A ducts were cultured in basal assay medium, consisting of phenol red-free DMEM/F12 (1:1, Gibco), 2% horse serum (Invitrogen), 5 ng/ml rhEGF (Peprotech), 0.5 μg/ml hydrocortisone (Sigma), 100 ng/ml cholera toxin (Sigma), 10 μg/ml insulin (Sigma) and 1% penicillin/streptomycin (Life Technologies). Human dermal microvascular endothelial cells (hMVEC-Ds, Lonza) were cultured in EGM2 medium (Lonza) supplemented with an MV2 bullet kit (Lonza). For co-culture in microfluidic devices, hMVECs were cultured in a reduced EGM2-MV, composed of the complete media kit with the exception of 0.5% serum and 0 ng/ml VEGF. HEK-293T cells (Clonetech) were grown in high glucose DMEM (Hyclone) supplemented with 10% fetal bovine serum (Hyclone) and 1% penicillin/streptomycin (Life Technologies). All cells were cultured at 37 °C and 5% CO$_2$ in a humidified incubator. Cell lines were tested for mycoplasma contamination using MycoAlert Mycoplasma Detection Kit (Lonza).

**Antibodies and reagents**. Anti-VE-cadherin (F-8, 1 μg/ml) and DAPI were from Santa Cruz Biotechnology. Anti-E-cadherin (ab1416, 1 μg/ml), anti-Ki67 (ab15580, 0.25 μg/ml), and anti-β tubulin (ab6046, 0.25 μg/ml) were from Abcam. Anti-GM130 (clone 35, 2 μg/ml) was from BD Biosciences. Anti-VEGFA (VG-1, 2 μg/ml), anti-ZO-1 (40-2200, 1 μg/ml), rhodamine phalloidin (1 μg/ml), 70 kDa FITC-dextran and AlexaFluor 647 conjugated goat secondary antibodies were from Life Technologies. Anti-α6 integrin (MA6, 1 μg/ml) was from Millipore. Anti-HA (6E2, 0.5 μg/ml), anti-GFP (D5.1, 0.5 μg/ml), and anti-GAPDH (D16H11, 0.25 μg/ml) were from Cell Signaling Technologies. HRP-conjugated donkey anti-mouse and rabbit IgG secondary antibodies (1: 0.25 μg/ml) were purchased from Fitzgerald. Recombinant human IL-6 protein (7270-IL, 200 ng/ml), recombinant human TGFβ1 protein (240-B, 5 ng/ml), recombinant human FGF2 protein 2 (33-FB, 3 nM), anti-IL-6 (6708, 1 μg/ml), anti-IL-6Rα (MAB227, 0.5 μg/ml), and Proteome Profiler Human Cytokine Array Kit were purchased from R&D Systems. Semaxanib was purchased from Selleckchem.

**Cloning and retroviral infection**. MSCV-IRES-GFP, MSCV-VEGFA-IRES-GFP, empty pBABE-Puro retroviral vector (EV) and pBABE vectors carrying wild-type *HER2/ERBB2* (Addgene plasmid #40978) or *PIK3CA(H1047R)* (Addgene plasmid #12524) were co-transfected independently with pUMVC (Addgene plasmid #8449) and VSV-G (Addgene plasmid #8454) retroviral packaging plasmids into HEK-293T cells using calcium phosphate transfection. Forty-eight hours after transfection, the viral supernatant was collected, concentrated using PEG-IT viral precipitator (SBI), and resuspended in PBS. The viral particles were then added to low passage (passages two to four) MCF10A cells in growth medium and incubated overnight. Forty-eight hours following transduction, MCF10A cells were plated into 6-well plates at a density of 100,000 cells per well and selected using 2 μg/ml puromycin (Gibco) for two days. Gene transduction and protein expression were validated by western blotting. Upon validation, established lines were immediately expanded and cryopreserved in several low passage aliquots.

**Device fabrication, seeding, and culture**. Microfluidic devices were fabricated using soft lithography (Supplementary Fig. 1a). Polydimethylsiloxane (PDMS, Sylgard 184, Dow-Corning) was mixed at 10:1 base:curing agent and cured overnight at 60 °C on a silicon master. The PDMS was cut from the silicon master, trimmed, and surface activated by plasma treatment for one minute. Devices were then bonded to glass and treated with 0.01% poly-L-lysine and 1% glutaraldehyde, washed in sterile water overnight, and sterilized with 70% ethanol for 30 min. Steel acupuncture needles (160 μm diameter, Seirin) were introduced into the device and devices were UV sterilized for 15 min. Rat tail collagen type I (BD Biosciences) solution was buffered with 10x reconstitution buffer (0.2 M HEPES, 0.02 M sodium bicarbonate) and 10X DMEM, titrated to a pH of 7.6 with 1 M NaOH, and brought to a final concentration of 2.5 mg/ml collagen I in cold PBS solution. 40 μl of collagen solution was injected into microfluidic devices and polymerized for 30 min at 37 °C to generate hydrogels with an elastic modulus of ~130 ± 39 Pa. Devices were washed with PBS and needles were removed to create a hollow 160 μm diameter central channel and duct channel in the collagen hydrogel. Fifty microliters of 2% GFR MG (~160 μg/ml, Corning) diluted in cold PBS was added to the duct port and passively absorbed onto the duct channel surface overnight at 4 °C. The following morning, devices were washed three times with ice cold PBS.

hMVECs were used between passages four and six. Cells were harvested with 0.05% Trypsin/EDTA and centrifuged at 200 × *g* for 5 min. Cells were resuspended at 0.5 × 10$^6$ cells/ml in EGM2-MV, and 80 μl of cell suspension was introduced into the central device channel to allow cells to adhere to collagen for 20 min before washing with growth medium. Once seeded, devices were cultured with EGM2-MV under perfusion using a laboratory rocker inside a tissue culture incubator for a minimum of 6 h (BenchRocker 2D, lowest speed setting, rocker angle 25°, maximal volumetric flow rate: ~1.06 μl/s, maximal shear stress: ~3.3 Dyn/cm$^2$)[21,22]. Wild-type or modified MCF10A cells were thawed, cultured in growth medium, and used between passages five to ten. Cells were harvested with 0.05% Trypsin/EDTA, neutralized with soybean trypsin inhibitor, and centrifuged at 200 x *g* for 5 min. Cells were resuspended at 0.2 × 10$^6$ cells/ml in basal assay medium, and 80 μl of cell

suspension was introduced into the ductal device port to allow cells to adhere to the Matrigel-coated top and bottom of the epithelial duct channel to a density of 75% (~1000 cells). For GFP/VEGF experiments, 100% modified cells were seeded. For the breast cancer mutation experiments, a mixture of 90% wild-type and 10% mutant cells were seeded. After seeding, the MCF10A suspension was replaced by an equal volume of basal assay medium. Devices were cultured overnight at 37 °C, 5% $CO_2$ with perfusion along the endothelial channel using a laboratory rocker[21,22]. One day after seeding, the medium in the endothelial channel was changed to reduced EGM2-MV. Culture medium was changed daily, adding basal assay medium to the epithelial port and reduced EGM2-MV to the central endothelial ports. Devices were monitored by brightfield imaging and were assayed between 4 and 5 days of co-culture. For the introduction of morphogen gradients, no endothelial cells were seeded into the central channel and basal assay medium was added to the central ports. Upon reaching duct confluence, the assay medium in the central channel was supplemented with either DMSO vehicle, 3 nM FGF2 or 5 ng/ml TGFβ1. Medium was changed daily, and devices were fixed after 3 days for analysis. A finite element model to characterize the dynamics of mass transport from the epithelial compartment in our model is described in the Supplementary Methods.

**Measurement of collagen hydrogel stiffness**. Collagen hydrogels were prepared as outlined above. To determine the stiffness of these collagen matrices, nanoindentation characterization was performed using a Piuma Nanoindenter system (Optics 11, Westwood, MA). The spherical indentation probe has a diameter of ~100 μm with cantilever spring constant (k ~0.5 N/m). Samples were immersed in PBS and measurements were performed at indentation depth of 10 μm and displacement speed of 5 μm/s at room temperature. The Young's moduli of these samples were calculated using the built-in Piuma software by fitting force-indentation curves to the established Hertzian contact mechanics model, assuming a Poisson ratio of 0.5 for incompressible materials. Data analysis used the Optics11 DataViewer. Six measurements were obtained from two hydrogels averaged a Young's Modulus of 130 ± 39 Pa.

**Immunofluorescence**. To fix devices for morphological analysis, devices were perfused with a solution of 4% paraformaldehyde (Electron Microscopy Sciences) in DMEM/F12 medium for 30 min at 37 °C. Devices were then washed three times with PBS and permeabilized in 0.25% Triton-X (Sigma) for 30 min at room temperature on a rocker. The devices were blocked using a 3% BSA solution added to all ports for 2 h at room temperature or overnight at 4 °C on a rocker. The devices were incubated with primary antibodies at their stated dilutions in 3% BSA overnight on a rocker at 4 °C. The devices were then washed three times in PBS and stained overnight on a rocker at 4 °C with secondary antibody in a 3% BSA solution. Samples were then washed three times with PBS. The devices were imaged on a Leica SP8 laser scanning confocal microscope (Leica Microsystems) with Leica HC FLUOTAR L 25×/0.95 W VISIR or a Leica HCX APO L 10×/0.30 W U-V-I controlled by LASX software. All permeability analysis imaging was collect using a Yokogawa CSU021/Zeiss Axiovert 200 M inverted spinning disk microscope using a Zeiss 10X Plan-APOCHROMAT objective controlled by Metamorph software. Composite images were acquired in spatial sequence using equal laser intensity and detector gain and manually stitched in ImageJ. Fluorescence images were adjusted for contrast and brightness using ImageJ.

**Conditioned medium assays**. Conditioned media were gathered from cultured MCF10A expressing EV, ErbB2$^{amp}$, or PI3Kα$^{H1047R}$. Protein concentration was normalized between samples using a colorimetric BCA assay (Pierce) and medium was concentrated using Amicon Ultra 0.5 ml 3 kDa filters (MilliporeSigma). For functional conditioned media experiments, 80 μl of basal assay medium supplemented with 200 ng/ml of rhIL-6 (R&D Systems), conditioned EV medium, conditioned PI3Kα$^{H1047R}$ medium, or conditioned EV medium + 200 ng/ml rhIL-6 was added from the duct port to stable endothelial vessels two days after seeding and cultured overnight before fixation or permeability analysis. For secretome analysis, conditioned media were treated with 1:100 Halt Protease and Phosphatase Inhibitor (Thermo), snap frozen in liquid nitrogen, and stored at −80 °C prior to western blot analysis. A Proteome Profiler Cytokine Array (R&D systems) was used to profile secreted cytokine content according to manufacturer instructions. To compare between membranes, densiometric intensities were normalized and compared to EV secretion using Eq. (1)

$$\frac{\text{Cytokine}(i)_{\text{Mutant}} - \text{Background}_{\text{Mutant}}}{\text{Cytokine}(i)_{\text{EV}} - \text{Background}_{\text{EV}}} \times \frac{\text{Protein Content Reference}_{\text{EV}}}{\text{Protein Content Reference}_{\text{Mutant}}}. \quad (1)$$

**Image processing and analysis**. To quantify Golgi polarity, a line was drawn in ImageJ from the center of the nucleus to the centroid of GM130 localization. The angle of this line was calculated in relation to the apical–basal nuclear axis, yielding the angle of the nuclear-GM130 axis. To quantify number of Ki67+ nuclei, confocal projections of epithelial ducts stained with Ki67 and DAPI were thresholded and masked for nuclei (DAPI) and the total number of Ki67+ nuclei were quantified via MATLAB. The fractional percentage was calculated by dividing the co-stained nuclei by the total number of nuclei. To quantify the diameter of

endothelial vessels, the width of phase-contrast images of endothelial vessels was measured at three distinct locations along the vessel in each image. The mean of the measurements was reported as the diameter of the vessel. To quantify vessel sprouts, sprouts along both sides of the vessel were manually counted in ImageJ.

To quantify VE-cadherin in endothelial vessels, images were processed in ImageJ. Images were background subtracted, binarized using Otsu's method, and quantified via integrated pixel density[22]. To quantify junctional actin, line profiles were drawn through the short axis of the cell, passing through the nucleus. Phalloidin signal intensity was measured along this line in three to four cells per vessel for three distinct vessels. Intensity values and length were normalized to their respective maxima. Using MATLAB, the total area under each intensity profile was calculated, as was the area contained within the 10% of the total length from each edge of the cell (0–10% and 90–100% of total length). Actin junctional localization was calculated by Eq. (2)

$$\text{Actin Junctional Localization} = \frac{A_{0-10\%} + A_{90-100\%}}{A_{\text{Total}}}. \quad (2)$$

**Permeability analysis**. Devices were imaged on a spinning disk confocal microscope in a live cell imaging chamber. Each of the endothelial reservoirs was drained and an equal volume of 2% 70 kDa fluorescent dextran in EGM2-MV medium was added to a single endothelial reservoir. The vessel was imaged every 10 s for 5 min at the mid-vessel plane. Time-lapse microscopy was used to measure the flux of dextran into the collagen gel, and the resulting diffusion profile was fitted to a dynamic mass-conservation equation[21,22], with the diffusive-permeability coefficient ($P_D$) defined by Eq. (3)

$$J = P_D(c_{\text{vessel}} - c_{\text{ECM}}) \quad (3)$$

where $J$ is the mass flux of dextran, $c_{\text{vessel}}$ is the concentration of dextran in the vessel, and $c_{\text{ECM}}$ is the concentration of dextran in the perivascular ECM. The devices were fixed for further analysis immediately after permeability testing.

**Statistical analysis**. Sample sizes and $P$ values are reported in each of the figure legends and Source Data. All statistical analysis was performed in GraphPad Prism 6. Unless otherwise specified, multigroup analysis was performed using a one-way analysis of variance (ANOVA) with a Bonferroni post-hoc test to report adjusted $P$ values, while dual group analysis was performed using an two-tailed, unpaired Student's $t$ test. Experiments were not randomized, nor were experimenters blinded to the data during analysis. With the exception of the cytokine array in Supplementary Fig. 7, all micrographs are representative of at least three independent experiments.

**Reporting summary**. Further information on research design is available in the Nature Research Reporting Summary linked to this article.

## Data availability

Source Data are provided for each figure. The source images for the western blots in the figures are available in Supplementary Fig. 8. Additional data that support the findings of this study are available from the corresponding author upon reasonable request. Source data are provided with this paper.

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

## Acknowledgements

This work was supported by grants from the NIH (R01HL147585, R01EB00262, R01EB008396), the National Science Foundation Center for Engineering Mechan-oBiology (CMMI15-48571), and the Biological Design Center at Boston University. M.L.K. acknowledges support from the NIH (K99CA226366, R00CA226366). T.K. acknowledges support from an EMBO Long-Term fellowship (ALTF 552-2016).

## Author contributions

M.L.K., W.J.P., and C.S.C. conceived the study and designed experiments. M.L.K., W.J.P., M.K.W., K.A.G., S.K., L.L., and T.K. performed all experiments and data analysis with critical input from A.I.M., M.L.K., W.J.P., and C.S.C. wrote the manuscript with input from all authors.

## Competing interests

The authors declare no competing interests.
