## [Peer Review File · Nature Communications]

Reviewers' Comments:

Reviewer #1:

Remarks to the Author:

In this manuscript, the authors produce a coculture platform of breast ducts and vessels including microfluidics with the intent of studying the paracrine effect of the ductal epithelial cells on vessel physiology and, for part of work, under epithelial genetic conditions known to be involved in pathogenesis (notably HER2/ERBB2 amplification and Pi3K mutation associated with breast cancer via the HER2/Pi3K/Akt pathway). Modulation of morphogenesis is induced either via gradients of growth factors (towards the ductal epithelium) or via overexpression of VEGF by epithelial cells (with an impact towards the endothelium).

The fact that the ducts covered with breast epithelial cells respond to growth factors by modifying their morphogenesis is a very interesting aspect of the potential power of this platform to bring important biological answers. This possibility is allowed by the means by which the channels are created within a malleable collagen I matrix. Another interesting aspect of the work is the effect of epithelial cells with an altered HER2/Pi3K/Akt pathway on the vascular endothelium, with differential impact on permeability depending on the type of pathway alteration that corroborates different levels of alterations also observed in breast cancers. Overall, this is a very interesting platform that is worthwhile for mechanistic studies and with which the authors already demonstrate new results regarding paracrine effects between epithelial and endothelial cells.

However, some of the conclusions made by the authors are not readily supported by the results shown, especially when matters like phenotypically normal differentiation and controlled environment are being discussed. A similar issue exists for the interpretation of the results with epithelial cells engineered to harbor an altered HER2/Pi3K/Akt pathway, for which more convincing evidence that they reproduce pathological stages (like tumorigenic progression) as observed in vivo would be necessary. Indeed, when we claim 3D cell culture models as more physiologically relevant, it usually requires demonstrating in depth similarities with relevant aspects of in vivo conditions.

Major comments:

Comment 1: In the introduction the authors should make clear why they only used epithelial cells and endothelial cells and not myoepithelial cells and fibroblasts as these might further influence the system and its response to paracrine factors. This is well discussed in the discussion section, but it seems important to justify focusing on only two cell types already in the introduction.

Comment 2: In a few places in the manuscript the authors claim that their system permits a controlled environment. This is not totally the case. Using Matrigel to cover the inside of the channels does not constitute a "controlled" microenvironment since the exact composition of Matrigel is not known usually and this composition varies with batches. Laminin coating would be a more controlled system instead of Matrigel. Did the authors try using this ECM component to provide polarity signaling? Moreover, the cell culture conditions make use of serum that is also a noncontrolled parameter of the microenvironment. Therefore, the authors should revisit their claim of a controlled environment.

Comment 3: The recapitulation of full polarity also seems an overstatement. It has been repeatedly shown by others that MCF10A do not polarize properly (this is why only a Golgi marker is used with these cells, but it is not a marker of apical polarity; tight junctions are markers of apical polarity; since most of the MCF10A lines available do not have tight junctions, we cannot call the hole in the center of the epithelial structure a lumen (the definition of a lumen in cell biology requires tight junctions)). The text for the MCF10A cells should be modified to reflect the fact that they only undergo partial polarization. It might have been possible to show full polarity with an epithelium formed by primary epithelial cells, but it is not shown. Of note, maturation over

4-5 days is a relatively short amount of time, but it is usually enough for basal polarity to be established; however, it is usually not enough for apical polarity (that could be normally obtained with primary cells) to be in place.

The lack of proper apical polarity would influence the way cells react to paracrine factors most likely and thus, it is important to be exact with the differentiation stage. In fact, it is likely that FGF can lead to cell multilayering because there is no apical polarity, since it was shown by others that if apical polarity is present, not even a strong growth factor can lead to proliferation. It is fine to keep the model as is, but the facts regarding the presence or lack of presence of complete polarity and the consequences for the response to growth factors need to be corrected and discussed appropriately.

There is no evidence of tight junction localized apically for the endothelium (ZO-1 is at cell-cell contacts, but there is no reconstruction of confocal images showing the apical localization of ZO-1).

Comment 4: From the image shown in Supplementary Figure 2A, it is not obvious with the primary cells that there is a dual layer with myoepithelial cells and luminal cells; only one layer of cells is seen by this reviewer, with the exception of a couple of cells inside the 'lumen' against the continuous monolayer of cells. Since from the image only one layer of cells is observed, it could very well be that the cells have myoepithelial features at their basal side as it is the case with many of the breast epithelial cells used 3D culture. Convincing pictures should be shown to support the presence of myoepithelial and luminal epithelial cells with additional markers tested, or the text should be modified to better reflect the results that are shown currently.

Comment 5: For figure 2C, it is confusing that the authors conclude that VEGF is not acting solely via EGFR. It is not clear why they can make this conclusion as the lack of total recovery with SEM treatment might be due to other reasons (e.g., the VEGF receptors may not all be saturated, or that there might already be internal changes that have occurred (if the inhibitor was used after VEGF exposure started)). More explanations should be given to support this statement.

Comment 6: Regarding the tumor progression effect. The text in the manuscript refers to both mammary development and pathogenesis, especially for the latter, tumor progression. Yet, only non-neoplastic cells are used in the experiments. It is difficult to reconcile the observations reported with actual cancer phenotypes unless cultures are prepared for pathological analysis in a way similar to that used for cancers (i.e., paraffin embedding with hematoxylin-eosin staining and assessment by a pathologist expert in breast cancer who could look at parameters of the cells usually considered for making a diagnosis of cancer). These results once obtained should be shown in the main figures.

Comment 7: The authors should revise the discussion section to carefully comment on the capabilities of the platform and remove overstatements (like the claim that the model mimics tumor progression, unless they include additional results that would readily confirm such claims). If they decide that they will not provide additional experiments to determine if they can keep these claims, then the discussion should be modified as should be the presentation of the results in order to optimally interpret the physiological relevance of the phenotypes that they are observing. In fact, the cells with altered HER2/Pi3K/Akt pathway may not be tumorigenic, but they may simply demonstrate capabilities to be invasive (as it occurs during normal branching morphogenesis for instance). The authors could also propose that although they may not have a tumor phenotype in the experiments that are reported in the manuscript, the morphogenetic changes that they are observing might explain why tumors with such genetic alterations have a distinct behavior at the invasive stage or might lead to lumen filling with cells. Of course, the impact of genetic alterations on mammary gland morphogenesis that might prime individuals for breast cancer is an interesting approach to consider and I am wondering if anything is known regarding the organization of the mammary gland of individuals who bear these genetic alterations.

Comment 8: More information is needed regarding some of the methods. Notably, how much Matrigel is used to cover the inside of the channels is not clearly explained and how much collagen I is used to fill the chamber as well as the stiffness used are missing pieces of information. Indeed, collagen I stiffness influences cell phenotype and the authors should use a Young's modulus corresponding to normal breast tissue. If the Young's modulus cannot be identified from the manufacturer's information, it should be measured (e.g., it is assumed to be ~800 Pa when measured by indentation on an unconstrained sample). If the stiffness is different than that of normal tissue stroma, the authors should interpret their results based on the existing difference. Indeed, it was shown by others that non-neoplastic cells can display an invasive phenotype upon loss of polarity only when the stiffness of the matrix used for the cell culture is reduced (for a recent example see Fostok et al, Cancers (Basel), 2019).

It is not clear how perfusion was maintained for the vessel. There should be information on the fluid flow (speed, etc. with the pump or the rocking speed if a pump was not used).

Minor comments:

- 1) There is no scale bar apparently on the image of supplementary Fig. 2b, although it is mentioned in the figure legend.
- 2) In the main text numbers less than 10 should be written out with letters unless they relate to a fundamental unit (e.g., g, ml, mm); this is not always correctly written in the text.
- 3) Normally it is ml instead of mL (although it seems that some journals accept the use of mL); the same comment can be made for μ l;
- 4) Figure 2 legend: it should be 10 μ M instead of 10 uM (the authors should make sure that they catch all instances of incorrect writing of units).
- 5) The authors should make sure that they also have a space between a number and its accompanying unit, as they have a few examples of numbers 'glued' to their unit in the text.
- 6) The proper way to write is medium (not media) when singular and related to cell culture (e.g., legend of figure 2 and many other places).
- 7) The last statement in the legend of figure 2 is not clear. Why using this medium shows that there is no loss of epithelial barrier?
- 8) In the first sentence (on line 2) of the results section on IL-6 the authors probably meant "invading epithelial cells" instead of "invading endothelial cells".
- 9) *In vitro* and *in vivo* are usually written in italics.
- 10) in the materials and methods, dilutions are indicated for the use of antibodies, although it is more appropriate to indicate the final concentration used when available, since the stock concentration might vary from one batch to another, and the concentration used is also indicative of the strength of the antibody.
- 11) It would have been interesting to know what happens if the growth factors are coming from within the vascular channel lined with differentiated endothelial cells compared to factors coming from an acellular vascular channel.
- 12) There are a few misspelled words or added words or missing words in the text that require the attention of the authors.

Reviewer #2:

Remarks to the Author:

In their manuscript Kutys et al. develop a novel microfluidic platform that allows the 3D co-culture of mammary epithelial cells and endothelial cells. Until now it is the first type of device where normal (not tumourigenic) mammary epithelial cells and endothelial cells are co-cultured as ducts, both resembling their in vivo 3D morphology and architecture. Furthermore the authors show that the two cell compartments are not isolated but they can cross-communicate via a paracrine signalling. The authors propose their 3D device as a novel system that helps dissecting the cross talk between epithelial and stromal cells in mammary gland and uncovering cell phenotypes. The study is well designed, with clear objectives. The manuscript is well written, with good quality images. The reader really appreciates the cartoons included in many figures, which help the understanding of device set-up.

Anyway I found that some points of the study would need to be further elucidated, in order to attest the relevance of the 3D device developed by Kutys et al. in the field of mammary gland research.

MAJOR POINTS:

1. Figure 1. I would recommend the authors to include an epithelial polarity marker like ZO1 and/or sialomucin to unequivocally demonstrate that epithelial cells have an apical polarized membrane domain.

Debnath et al (Cell, 2001) showed that apoptosis plays a key role in lumen formation in MCF10A cells acini cultured in Matrigel. I would analyse apoptosis (by caspase 3 staining, for example) in mammary duct in the 3D device as well. It would be interesting the authors could compare their own device with already published platforms experimentally, i.e. see if the 3D mammary duct recapitulates or not what previously shown in 3D acinus.

2. The authors highlight the paracrine signalling active between vascular and mammary duct channels. While this signalling is well characterized in presence of genetic alterations of the mammary component, it is not sufficiently illustrated in normal conditions.

It is not clear if the co-culture of normal endothelial and normal mammary ducts stimulates and contributes to the correct maturation of both compartments, presumably via the same paracrine signalling active in pathological-mimicking conditions. The authors describe the architecture of vascular and mammary ducts at 1 week of co-culture, no study is shown at earlier time points. In the legend of Supplementary Figure 1a, the authors state "the epithelium matures over 4-5 days for stable co-culture". In Supplementary Figure 1b the phase contrast images apparently show lateral sprouting/branching. If the co-culture allows mammary branching, please show a quantification of branching at different days of co-culture (for example at day 1, 3/4 and 7). If the co-culture allows mammary cells to reach the correct polarized architecture, please show staining for polarity markers and apoptosis/proliferation on mammary ducts at different days of co-culture (for example at day 1, 3/4 and 7).

In the morphogen gradient studies mammary cells apparently assemble as a confluent monolayer (see Fig. 1d, vehicle) in the absence of vascular cells. Please show a staining for polarity markers on mammary duct cultured alone, where no endothelial cells are seeded. That would help to clarify the value of the co-culture.

It is crucial for the authors to describe if and how the two cell compartments benefit from the co-culture in normal conditions.

3. Supplementary Figure 2a. The authors show an immunofluorescence staining on HMEC cells at 5 days of co-culture, however no study is mentioned in the manuscript regarding the co-culture of HMEC cells and endothelial cells. I would recommend the authors to include an assay where these

two cell types are co-cultured to test if the device could be valid with a second mammary cell line.

4. Page 7, line 10. The authors say "...and an invasive phenotype representative of epithelial-to-mesenchymal transition...". To support this statement the authors should perform and show a staining for mesenchymal markers (like Vimentin, Snail) on mammary duct exposed to TGF β 1 gradient. If not it would be better to change "representative" into "suggesting".

5. Figure 2a. It is not clear if the authors seed 100% VEGFA overexpressing cells. Please clarify.

6. Figure 2b. Please show quantification of vascular vessel diameter with VEGFA expressing ducts in which DMSO (VEGF + Veh) or Semaxanib (VEGF + Sem) are delivered.

7. Supplementary Figure 2c. Since VEGFA is a secreted protein, please show Western Blot on the supernatant (conditioned medium) of IRES-GFP and VEGF-IRES-GFP overexpressing MCF10A cells.

8. What is the phenotype of VEGFA overexpressing cells when seeded alone and when in co-culture? The authors focus on the vascular remodelling induced by VEGFA overexpressing cells. Does VEGFA overexpression lead to any morphological/polarity state change in MCF10A cells? I would perform a staining on VEGFA overexpressing cells when seeded alone and when co-cultured.

9. To unequivocally prove that vascular remodelling is induced by VEGFA overexpressing cells via paracrine signalling I would recommend culturing vascular vessels with conditioned medium from GFP expressing ducts and VEGFA expressing ducts and perform quantification of vessel diameter and vessel sprouts in each condition.

10. Page 10, line 9. The authors state "invasion of PI3 α H1047R ducts occurred more rapidly than Erbb2amp and had distinct mesenchymal morphology". First, if the authors refer to a higher speed in the invasion process I would recommend to perform a time lapse analysis. Second, did the authors perform any staining for mesenchymal markers?

11. Page 10, line 17. Please show quantification of neo-angiogenic sprouting.

12. The authors mixed mutant cells with wtype cells in a ratio 1:10. How mutant and wtype cells re-organize in the mosaic ducts? Are PI3 α H1047R and Erbb2amp cells driving invasion? I would perform an anti-HA staining on PI3 α H1047R and Erbb2amp ducts.

13. Could PI3 α H1047R and Erbb2amp ducts phenotypes be dependent on signalling coming from vascular vessel? To unravel this I would suggest the authors to culture PI3 α H1047R and Erbb2amp ducts alone and perform staining for actin.

14. Figure 4a. Please show Western Blot for IL-6 and IL-6Ra on basal media (BM).

MINOR POINTS:

1. Figure 1, legend. It would be better to clearly specify in the introductory head that all the IF images shown in this figure refer to MCF10A cells at 1 week of co-culture.

2. Figure 1b. Please correct "nuceli" into "nuclei"

3. Supplementary Figure 2a. I would put it in Supplementary Figure 1, since it is related to the device bioengineering and not to the morphogen gradient assay.

4. Figures 2a and 3b. Please describe in the section "Material and Methods" how VEGFA overexpressing MCF10A cells and PI3 α H1047R/Erbb2amp MCF10A cells were maintained before

using the device and how they were seeded in the device (number of cells, culture medium). In particular, in the text (page 10, line 5) it is said mutant cells were mixed 1:10 with wild type cells. Please describe it in "Material and Methods".

5. Page 10, line 5. Please explain why mosaic ducts were generated. I would modify the cartoon in Figure 3a accordingly, i.e. I would use two different colours to indicate mutant and wtype cells in mosaic mammary duct.

Reviewer #3:

Remarks to the Author:

This study focuses on the development and characterization of a microphysiological system to investigate the crosstalk between mammary epithelial cells and their neighboring vasculature. The system consists of an endothelialized channel adjacent to a human mammary duct. Following characterization, the authors use this platform to study the effect of epithelial cell HER2/ERBB2 amplification or PIK3CA(H1047R) mutation on vascular sprouting and permeability. They suggest that IL-6 is a key player in driving endothelial cell dysfunction due to PI3K mutated mammary epithelial cells. Strengths of the manuscript include the uniqueness and simplicity of the microfluidic system and that the manuscript is clearly written. However, in its current form the model is not sufficiently well described and the biological studies are missing control conditions to substantiate the conclusions regarding the role of IL-6 in the observed vascular changes.

Major comments:

1. The best part of the model system is that it allows studying specific aspects of the reciprocal crosstalk between epithelial cells and the vasculature under in vivo-like culture conditions while permitting isolated manipulations. Clearly, there is a tremendous need for such platforms, but the paper could do a much better job at motivating and characterizing the design of the model and demonstrate its relevance:

a. While there is some data describing transport characteristics from the vascular channel, a more complete characterization would be helpful. For example, is gradient generation stable over the culture period? How fast does the gradient establish? Also, what is the motivation for separating the vascular channel and duct 500 μm apart (Krogh length is 100-200 μm). Does this distance affect the gradient established between both systems?

b. Figure 2 focuses on the effect of a gradient from the duct channel on the function of the vascular channel. Therefore, it may be helpful to include transport characterization of the duct channel as well.

c. Is there an explanation for why mutant MCF10As were seeded at a 1:10 ratio with wild type cells for experiments in Figure 3?

d. Figure 4C repeats the diffusive permeability measurements that were performed in 3C, but the magnitudes are reduced by an order of 10. Is there an explanation for this? Perhaps the difference in exposure time to conditioned media (one week co-culture vs overnight treatment with conditioned media), which should be tested with a control experiment.

e. Most of the work was performed with MCF10A, but some experiments utilized primary human mammary epithelial cells (HMECs). This comparison is valuable, but requires further explanation since HMECs appeared to form glands differently. For example, glands seemed smaller, but a direct comparison is difficult since the size of the scale bar in Suppl. Fig. S2 is missing.

2. If approached from a more biological rather than engineering design perspective, the experiments are not well-integrated and important control conditions are missing:

a. For example, Fig. 1 shows the effect of morphogen gradients on the epithelial cells (i.e., independent of endothelial cells). Fig. 2 tests how VEGF secretion of epithelial cells affects the endothelial cells, and Figs. 3 and 4 study the effect of epithelial cells dysfunction on the vascular cells independent of VEGF. However, what is missing is how all of these aspects may be interrelated. For example, the authors show that IL-6 alters vascular permeability. These changes

not only affect morphogen delivery from the vascular channel, but also suggest endothelial cell phenotypic changes that could independently affect epithelial cell invasion.

b. Did treatment with Semaxanib affect the duct channel in any way? VEGF signaling has been shown to affect breast cancer behavior in an autocrine/paracrine fashion (Guo S. et al., 2010)

c. A study that involves blocking IL-6 signaling in conditioned media would further solidify the claim that vascular dysfunction is driven by IL-6 secretion.

Minor comments:

1. It would be helpful to include color legends within the figures (like in some panels of Figure 1b) rather than in the figure caption.

2. Figure 1d, was the effect of growth factor gradients assessed in the presence of an endothelialized vascular channel in addition to an acellular vascular channel? The presence of endothelial cells could alter gradient generation or responses to growth factors.

3. The use of reference 25 seems out of place in the text. The relevance of VEGF-A to pregnancy is not investigated nor referred to elsewhere in the manuscript.

4. Figure 2D, for clarification pls include arrows pointing to sprouts that were quantified.

5. For the morphological differences seen in mammary epithelial cells due to ErbB2 and PI3Ka overexpression, the quantification of cell aspect ratio, invasive area, and single/multicellular invasion in the supplementary might fit better in the main body.

6. For supplementary Figure 4, there does not appear to be any description of sample sizes or number of tests performed.

7. Please explain why Figs. 3 and 4 focused on actin rather than CD31 or VE-Cadherin quantification.

8. To demonstrate luminal filling, it would be helpful to include a cross-sectional view in addition to the current maximum intensity projections. Also, a quantification of the data shown in Fig. 1d would be valuable to indicate reproducibility of the findings.

9. Please introduce all abbreviations at first mentioning.

Referee #1 (Remarks to the Author and our responses):

In this manuscript, the authors produce a coculture platform of breast ducts and vessels including microfluidics with the intent of studying the paracrine effect of the ductal epithelial cells on vessel physiology and, for part of work, under epithelial genetic conditions known to be involved in pathogenesis (notably HER2/ERBB2 amplification and Pi3K mutation associated with breast cancer via the HER2/Pi3K/Akt pathway). Modulation of morphogenesis is induced either via gradients of growth factors (towards the ductal epithelium) or via overexpression of VEGF by epithelial cells (with an impact towards the endothelium).

The fact that the ducts covered with breast epithelial cells respond to growth factors by modifying their morphogenesis is a very interesting aspect of the potential power of this platform to bring important biological answers. This possibility is allowed by the means by which the channels are created within a malleable collagen I matrix. Another interesting aspect of the work is the effect of epithelial cells with an altered HER2/Pi3K/Akt pathway on the vascular endothelium, with differential impact on permeability depending on the type of pathway alteration that corroborates different levels of alterations also observed in breast cancers. Overall, this is a very interesting platform that is worthwhile for mechanistic studies and with which the authors already demonstrate new results regarding paracrine effects between epithelial and endothelial cells.

However, some of the conclusions made by the authors are not readily supported by the results shown, especially when matters like phenotypically normal differentiation and controlled environment are being discussed. A similar issue exists for the interpretation of the results with epithelial cells engineered to harbor an altered HER2/Pi3K/Akt pathway, for which more convincing evidence that they reproduce pathological stages (like tumorigenic progression) as observed in vivo would be necessary. Indeed, when we claim 3D cell culture models as more physiologically relevant, it usually requires demonstrating in depth similarities with relevant aspects of in vivo conditions.

We would like to thank this reviewer for their positive assessment and constructive suggestions for the work. As detailed below, we were able to complete several additional experiments and make numerous revisions to the manuscript that we believe largely address the primary concerns raised. Unfortunately, due to COVID-19, there were a few studies we were not able to complete, but none of these substantially alter the primary takeaways of the work. See below for our responses to specific points raised.

Major comments:

Comment 1: In the introduction the authors should make clear why they only used epithelial cells and endothelial cells and not myoepithelial cells and fibroblasts as these might further influence the system and its response to paracrine factors. This is well discussed in the discussion section, but it seems important to justify focusing on only two cell types already in the introduction.

We appreciate the opportunity for clarifying our goals in the introduction, and to appropriately acknowledge the importance of other cell types that were not a focus of our model. In addition to the referenced Discussion, we have updated the Introduction to reflect this point and to include the additional clarification: "Here, we sought to develop a new experimental system capable of recapitulating and dissecting diverse 3D mammary morphogenic processes and complex paracrine interactions amongst multiple cell types.

To enable this longer-term vision, in this study we focused first on building and studying the interactions between two principle tissue structures of the mammary gland – a biomimetic human mammary epithelial duct cultured in proximity to an endothelialized vessel.”

Comment 2: In a few places in the manuscript the authors claim that their system permits a controlled environment. This is not totally the case. Using Matrigel to cover the inside of the channels does not constitute a “controlled” microenvironment since the exact composition of Matrigel is not known usually and this composition varies with batches. Laminin coating would be a more controlled system instead of Matrigel. Did the authors try using this ECM component to provide polarity signaling? Moreover, the cell culture conditions make use of serum that is also a noncontrolled parameter of the microenvironment. Therefore, the authors should revisit their claim of a controlled environment.

We apologize for the misunderstanding. Our intent was to suggest that our top-down fabrication of tissues provides newfound control over the physical architecture and arrangement of epithelial ducts and vessels, as compared to, for example, traditional models where spherical mammary acini form spontaneously. We agree with the reviewer that despite capabilities to provide control over some parameters that were not possible in most previous models, the use of growth factor reduced (GFR) Matrigel and serum remain as undefined inputs to the model. Previously, we had observed that naked (uncoated) collagen type I gels were not able to support stable duct formation, and that GFR Matrigel coating was sufficient. To address the variability and lack of control inherent in the use of Matrigel in light of the reviewer comment, we explored the use of purified basement membrane proteins as polarization stimuli. Interestingly, we found that coating with collagen IV or laminin alone was unable to support the formation of a stable/non-invasive mammary epithelium (Supplementary Figure 1b). While we had additional ECM proteins and combinations planned, we unfortunately were not able to complete those studies, but nonetheless this result better motivates the use of GFR Matrigel in our system and provides a roadmap to use defined ECMs to characterize the necessary molecular requirements for mammary epithelial polarization and behavior, which now will be explored in future work. We have added these additional data to the revised manuscript.

Comment 3: The recapitulation of full polarity also seems an overstatement. It has been repeatedly shown by others that MCF10A do not polarize properly (this is why only a Golgi marker is used with these cells, but it is not a marker of apical polarity; tight junctions are markers of apical polarity; since most of the MCF10A lines available do not have tight junctions, we cannot call the hole in the center of the epithelial structure a lumen (the definition of a lumen in cell biology requires tight junctions)). The text for the MCF10A cells should be modified to reflect the fact that they only undergo partial polarization. It might have been possible to show full polarity with an epithelium formed by primary epithelial cells, but it is not shown. Of note, maturation over 4-5 days is a relatively short amount of time, but it is usually enough for basal polarity to be established; however, it is usually not enough for apical polarity (that could be normally obtained with primary cells) to be in place.

We thank the reviewer for raising this important point. We agree that MCF10A are known to have immature apical organization and in our system these cells similarly failed to achieve a mature apical domain across all cells of the ductal epithelium. To better understand the molecular processes and cellular restructuring that occur during mammary duct stabilization in our model, we have conducted a time course of ductal assembly. We observed that GM130 reorganized from the basal to luminal surface during duct assembly.

We further observed that apoptosis, indicated by cleaved caspase 3, was important for luminal clearing of the subset of cells in the center of the lumen and lacking an ECM interface (Supplementary Figure 2a, 2b). We also quantified restructuring of cell morphology over this period (Supplementary Figure 2c). We did investigate whether the architecture and assembly mechanism of our ducts might promote ZO-1 or Mucin1 apical localization. Consistent with existing literature, MCF10A ducts displayed limited ZO-1 expression or apical structures. Mucin 1 displayed basal to apical restructuring cells (below); however, the expression of Mucin 1 was limited to a small fraction of cells. So, while the cells are able to polarize to distinguish apical and basolateral regions (as evidenced by GM130, $\alpha 6$ integrin, and E-cadherin), they do not develop mature apical domains and associated tight junctions. At the reviewer's suggestion, we have updated the text to reflect these points.

While certain luminal structures form concomitantly with the organization of tight junctions, we are employing the term as used broadly in anatomy to define the inside space of a tubular compartment lined by cells. Furthermore, in previous work the term “lumen” is commonly used when referring to the central, hollow region of MCF10A acini, despite the absence of tight junctions (Debnath et al. Cell. 2002; Leung, Nature, 2012), supporting our use of the term. However, to avoid confusion, we have provided additional language to clarify this use.

The lack of proper apical polarity would influence the way cells react to paracrine factors most likely and thus, it is important to be exact with the differentiation stage. In fact, it is likely that FGF can lead to cell multilayering because there is no apical polarity, since it was shown by others that if apical polarity is present, not even a strong growth factor can lead to proliferation. It is fine to keep the model as is, but the facts regarding the presence or lack of presence of complete polarity and the consequences for the response to growth factors need to be corrected and discussed appropriately.

We agree with the reviewer on the relationship between cell polarity and response to growth factor gradients. While polarized, bi-layered primary murine mammary organoids will undergo branching in response to FGF2 in 3D *in vitro* culture, it is important to consider that the growth factor responses of the engineered mammary ducts may be potentiated by the lack of tight junctions or a fully developed apical domain in our system. We have modified our Results to point out the extent of structural maturity of the MCF10A ducts, and in the Discussion we comment on the impact of this immaturity on the response to morphogen gradients.

There is no evidence of tight junction localized apically for the endothelium (ZO-1 is at cell-cell contacts, but there is no reconstruction of confocal images showing the apical localization of ZO-1).

We have conducted extensive characterization of our engineered microvascular systems (Polacheck & Kutys et al. Nature. 2017; Nguyen & Stapleton et al. PNAS. 2013) and demonstrated their utility as a platform to study endothelial barrier function. In this work, we demonstrate the junctional localization of ZO-1 (Figure 1) and the maintenance of vascular barrier function under normal conditions in our co-culture model, indicative of functional tight junctions. It is noted that microvascular endothelium form a very flat morphology in vessels, such that cell-cell boundaries are typically 200-500 nm in cross-section, and unlike in epithelia, those junctions are oblique and not vertical to the basement membrane. As such, fluorescence microscopy is not able to resolve positioning of junctions in the vertical direction. Transmission electron microscopy images suggest that tight junctions are not always apically situated except in specialized vasculature such as the blood-brain-barrier endothelium (for examples, see Wallez Y and Huber P, Endothelial adherens and tight junctions in vascular homeostasis, inflammation and angiogenesis, Biochimica et Biophysica Acta, Volume 1778, 2008, Pages 794-809; Haseloff R, Dithmer S, Winkler L, Wolbur H, Blasig I. Transmembrane proteins of the tight junctions at the blood–brain barrier: Structural and functional aspects. Seminars in cell & developmental biology. 38. (2014). 10.1016/j.semcd.2014.11.004).

Comment 4: From the image shown in Supplementary Figure 2A, it is not obvious with the primary cells that there is a dual layer with myoepithelial cells and luminal cells; only one layer of cells is seen by this reviewer, with the exception of a couple of cells inside the 'lumen' against the continuous monolayer of cells. Since from the image only one layer of cells is observed, it could very well be that the cells have myoepithelial features at their basal side as it is the case with many of the breast epithelial cells used 3D culture. Convincing pictures should be shown to support the presence of myoepithelial and luminal epithelial cells with additional markers tested, or the text should be modified to better reflect the results that are shown currently.

To address this concern, we conducted additional experiments to provide more detail on the formation and organization of HMEC ducts. However, we encountered unanticipated experimental reproducibility issues that we believe were due to phenotypic drift of our in-house primary HMECs. Ongoing and planned experiments with a new HMEC line unfortunately have not been straightforward and will require additional optimization of conditions to be incorporated confidently. While our preliminary data was supportive of the model's ability to support HMECs, we have removed the original Supplementary Figure 2a and the reference to HMEC in the Results to account for the lack of detailed characterization of HMEC in the system. The removal of these data does not influence the primary conclusions and new mechanisms in this manuscript, and our intention is to introduce HMECs in a follow-on study.

Comment 5: For figure 2C, it is confusing that the authors conclude that VEGF is not acting solely via VEGFR. It is not clear why they can make this conclusion as the lack of total recovery with SEM treatment might be due to other reasons (e.g., the VEGF receptors may not all be saturated, or that there might already be internal changes that have occurred (if the inhibitor was used after VEGF exposure started). More explanations should be given to support this statement.

The reviewer is correct, and we apologize for the misstatement. We have revised this statement to: “Semaxanib did not fully reverse VEGF-driven changes in endothelial cell morphology and actin organization (Fig. 2c), possibly because prolonged VEGF exposure produced cumulative endothelial restructuring that is not immediately reversible following inhibition of VEGFR2 signaling.”

Comment 6: Regarding the tumor progression effect. The text in the manuscript refers to both mammary development and pathogenesis, especially for the latter, tumor progression. Yet, only non-neoplastic cells are used in the experiments. It is difficult to reconcile the observations reported with actual cancer phenotypes unless cultures are prepared for pathological analysis in a way similar to that used for cancers (i.e., paraffin embedding with hematoxylin-eosin staining and assessment by a pathologist expert in breast cancer who could look at parameters of the cells usually considered for making a diagnosis of cancer). These results once obtained should be shown in the main figures.

We had set up many devices to address this important request during the final allotted month of revision. However, ongoing and planned experiments were halted due to COVID-related shutdowns. While we were unable to complete a careful pathological examination of the invasive behaviors in our system, we note that we have provided additional data characterizing their phenotypic differences in an expanded Supplementary Figure S5. We agree nonetheless that the suggested histologic studies would have provided additional evidence to compare to the traditional classification methods designed for biopsied human breast tissue. Lacking these data, we have significantly modified the text in line with the reviewer’s recommendation to remove implications that the approach might be an appropriate model for recapitulating cancer progression, but (as the reviewer helpfully points out in the next comment below) instead focus on its utility to isolate how specific alterations that occur in human breast cancer, when introduced independently, elicit morphogenic behaviors that may underlie certain tumor behaviors.

Comment 7: The authors should revise the discussion section to carefully comment on the capabilities of the platform and remove overstatements (like the claim that the model mimics tumor progression, unless they include additional results that would readily confirm such claims). If they decide that they will not provide additional experiments to determine if they can keep these claims, then the discussion should be modified as should be the presentation of the results in order to optimally interpret the physiological relevance of the phenotypes that they are observing. In fact, the cells with altered HER2/Pi3K/Akt pathway may not be tumorigenic, but they may simply demonstrate capabilities to be invasive (as it occurs during normal branching morphogenesis for instance). The authors could also propose that although they may not have a tumor phenotype in the experiments that are reported in the manuscript, the morphogenetic changes that they are observing might explain why tumors with such genetic alterations have a distinct behavior at the invasive stage or might lead to lumen filling with cells. Of course, the impact of genetic alterations on mammary gland morphogenesis that might prime individuals for breast cancer is an interesting approach to consider and I am wondering if anything is known regarding the organization of the mammary gland of individuals who bear these genetic alterations.

We appreciate this articulation by the reviewer and the idea that “the morphogenetic changes that they are observing might explain why tumors with such genetic alterations have a distinct behavior at the invasive stage or might lead to lumen filling with cells” is a key concluding principle we are intending to convey. At the reviewer’s suggestion, and in light of our inability to perform the required histological comparisons, we have removed

explicit claims that this model mimics tumor progression, and we have substantially revised the Introduction, Results, and Discussion to better reflect this key concluding principle as it relates to mammary epithelial invasive behaviors. We thank this reviewer for providing this constructive and insightful critique, which we believe places our work in a more appropriate context.

Comment 8: More information is needed regarding some of the methods. Notably, how much Matrigel is used to cover the inside of the channels is not clearly explained and how much collagen I is used to fill the chamber as well as the stiffness used are missing pieces of information. Indeed, collagen I stiffness influences cell phenotype and the authors should use a Young's modulus corresponding to normal breast tissue. If the Young's modulus cannot be identified from the manufacturer's information, it should be measured (e.g., it is assumed to be ~800 Pa when measured by indentation on an unconstrained sample). If the stiffness is different than that of normal tissue stroma, the authors should interpret their results based on the existing difference. Indeed, it was shown by others that non-neoplastic cells can display an invasive phenotype upon loss of polarity only when the stiffness of the matrix used for the cell culture is reduced (for a recent example see Fostok et al, Cancers (Basel), 2019).

We have added the specific volumes and concentrations of collagen I and Matrigel used in our device assembly protocol to the Methods. We performed nanoindentation characterization to determine the Young's moduli of our collagen hydrogels. We report an averaged modulus of approximately 130 ± 39 Pa as detailed in the Methods section. This value is in range of what has been reported by similar methods for normal mammary gland ducts *in vivo* ~400 Pa (Lopez et al. *Int Bio.* 2011).

An attractive aspect of the model is that it not only supports the formation of stable 3D mammary duct architectures in a physiologically compliant ECM, but it also allows the active remodeling of this matrix to support a diverse array of morphogenic behaviors in response to specific perturbations. While ECM stiffness can certainly influence epithelial polarity, how, and in what context this stiffness influences normal and tumor cell behavior is an area of active study in the field. Stiffness is only one of multiple physical parameters modulated in the provided reference. In that study, the authors vary dilutions of a bulk 3D Matrigel ECM (1:5, 1:10, 1:20) as a mechanism to modulate stiffness. These manipulations modulate ECM ligand concentration, porosity, and topography in addition to stiffness (which was not measured in the study), all of which could conceivably alter invasive behavior of non-neoplastic cells. Nonetheless, the important findings of that study provide important motivation for the development of platforms in which the role of these biophysical parameters can be controlled and studied in the context of cellular architecture and invasion. We now include this point in the Discussion of the revised manuscript.

It is not clear how perfusion was maintained for the vessel. There should be information on the fluid flow (speed, etc. with the pump or the rocking speed if a pump was not used).

Our original submission stated that a laboratory rocker was used to perfuse the vessel. We have added further details on rocker model, tilt angle, and speed, as well as calculated volumetric flow rate and shear stress based on our geometry (Methods and Supplemental Methods).

Minor comments:

1) *There is no scale bar apparently on the image of supplementary Fig. 2b, although it is mentioned in the figure legend.*

Thank you, we have added the scale bar to now Supplementary Figure 2e.

2) *In the main text numbers less than 10 should be written out with letters unless they relate to a fundamental unit (e.g., g, ml, mm); this is not always correctly written in the text.*

This has been updated in the revised manuscript.

3) *Normally it is ml instead of mL (although it seems that some journals accept the use of mL); the same comment can be made for μ l;*

This has been updated in the revised manuscript.

4) *Figure 2 legend: it should be 10 μ M instead of 10 uM (the authors should make sure that they catch all instances of incorrect writing of units).*

This has been updated in the revised manuscript.

5) *The authors should make sure that they also have a space between a number and its accompanying unit, as they have a few examples of numbers 'glued' to their unit in the text.*

This has been corrected in the revised manuscript.

6) *The proper way to write is medium (not media) when singular and related to cell culture (e.g., legend of figure 2 and many other places).*

This has been corrected in the revised submission.

7) *The last statement in the legend of figure 2 is not clear. Why using this medium shows that there is no loss of epithelial barrier?*

We apologize for this confusing statement in the Results section. We have revised this sentence for clarification, removing the reference to epithelial barrier function: "Importantly, treatment of vessels with basal mammary duct medium (BM) alone did not result in the vascular phenotypic changes (Fig. 2c) that were observed with the conditioned media experiments (Supp. Fig 4c). Together with the VEGF studies, these data prescribe the increased vascular diameter and sprouting phenotype specifically to VEGF paracrine signaling."

8) *In the first sentence (on line 2) of the results section on IL-6 the authors probably meant "invading epithelial cells" instead of "invading endothelial cells".*

Thank you. This is correct and has been updated.

9) *In vitro and in vivo are usually written in italics.*

This has been updated in the revised submission.

10) *in the materials and methods, dilutions are indicated for the use of antibodies, although it is more appropriate to indicate the final concentration used when available, since the stock*

concentration might vary from one batch to another, and the concentration used is also indicative of the strength of the antibody.

Antibody concentrations have been added to the Methods section.

11) It would have been interesting to know what happens if the growth factors are coming from within the vascular channel lined with differentiated endothelial cells compared to factors coming from an acellular vascular channel.

It's important to note that, as stated in the text, the application of growth factor gradients was to assess the capacity of our engineered ducts to undergo diverse morphogenic transitions from a stable tissue architecture as a proof of principle. We agree that it would be feasible in our setup to investigate the role of vascular barrier in modulating the ability of the blood compartment to control epithelial function, but this is one of many interesting questions that will be reserved for a separate study.

12) There are a few misspelled words or added words or missing words in the text that require the attention of the authors.

We believe that we have corrected all textual issues in our revised submission.

Referee #2 (Remarks to the Author and our responses):

In their manuscript Kutys et al. develop a novel microfluidic platform that allows the 3D co-culture of mammary epithelial cells and endothelial cells. Until now it is the first type of device where normal (not tumorigenic) mammary epithelial cells and endothelial cells are co-cultured as ducts, both resembling their in vivo 3D morphology and architecture. Furthermore the authors show that the two cell compartments are not isolated but they can cross-communicate via a paracrine signalling. The authors propose their 3D device as a novel system that helps dissecting the cross talk between epithelial and stromal cells in mammary gland and uncovering cell phenotypes.

The study is well designed, with clear objectives. The manuscript is well written, with good quality images. The reader really appreciates the cartoons included in many figures, which help the understanding of device set-up. Anyway I found that some points of the study would need to be further elucidated, in order to attest the relevance of the 3D device developed by Kutys et al. in the field of mammary gland research.

We thank this reviewer for their positive assessment of the study, and we provide additional revisions as a result of this reviewer's suggestions that we believe strengthen the work.

MAJOR POINTS:

1. Figure 1. I would recommend the authors to include an epithelial polarity marker like ZO1 and/or sialomucin to unequivocally demonstrate that epithelial cells have an apical polarized membrane domain. Debnath et al (Cell, 2001) showed that apoptosis plays a key role in lumen formation in MCF10A cells acini cultured in Matrigel. I would analyse apoptosis (by caspase 3 staining, for example) in mammary duct in the 3D device as well. It would be interesting the authors could compare their own device with already published platforms experimentally, i.e. see if the 3D mammary duct recapitulates or not what previously shown in 3D acinus.

To address the reviewer's comments and place the role of apoptosis in the context of prior work, we stained epithelial ducts with cleaved caspase 3 antibody to monitor apoptosis over the ductal maturation time course. Apoptosis as indicated by caspase 3 was present in early stages of ductal development and facilitates luminal clearing of the subset of cells that were not in contact with the basement membrane. These findings are analogous to the mechanism of lumenization described by Debnath and others. These new data are included in Supplementary Figure 2a, and we have added a description of these behaviors to the Results.

Despite MCF10A having been a critical resource for understanding studying normal and tumor-like mammary epithelial architecture and biology, it has been well documented by the Debnath and Brugge groups, among others, that this line has defects in assembling mature apical domains. A common marker for MCF10A apical-basal structuring is Golgi marker 130 (GM130), and we have conducted new experimentation that demonstrates reorganization of GM130 from the basal to luminal face over the time course of ductal assembly (Supplementary Figure 2b). This molecular reorganization corresponded with restructuring of cell morphology (Supplementary Figure 3c). We also assessed for changes in ZO-1 and Mucin 1. Consistent with existing literature, MCF10A ducts showed limited ZO-1 expression and localization to apical structures. Mucin 1 displayed basal to apical restructuring during the time course of assembly (below); however, the expression of Mucin 1 was limited to a small fraction of cells. We have updated our Results and Discussion to indicate that MCF10A achieve the same extent of polarity as described in other systems and discuss the limitations and implications for interpreting the responsiveness of such ducts to soluble morphogens.

2. The authors highlight the paracrine signalling active between vascular and mammary duct channels. While this signalling is well characterized in presence of genetic alterations of the mammary component, it is not sufficiently illustrated in normal conditions. It is not clear if the co-culture of normal endothelial and normal mammary ducts stimulates and contributes to the correct maturation of both compartments, presumably via the same paracrine signalling active in pathological-mimicking conditions. The authors describe the architecture of vascular and mammary ducts at 1 week of co-culture, no study is shown at earlier time points. In the legend of Supplementary Figure 1a, the authors state "the epithelium matures over 4-5 days for stable co-culture". In Supplementary Figure 1b the phase contrast images apparently show lateral sprouting/branching. If the co-culture allows mammary branching, please show a quantification of branching at different days of co-culture (for example at day 1, 3/4 and 7). If the co-culture

allows mammary cells to reach the correct polarized architecture, please show staining for polarity markers and apoptosis/proliferation on mammary ducts at different days of co-culture (for example at day 1, 3/4 and 7).

We apologize for not adequately addressing this point in our original submission. We observe no cooperative requirement for the culture of both endothelial and epithelial tissues for their stable development. We have extensively characterized the maturation of engineered microvascular systems in previous reports, and here we find the endothelial vessels proceed similarly whether in co-culture or cultured alone for conditioned media experiments (Figure 4). Similarly, we observed no dependence on the endothelium for the maturation of the mammary ducts. Figure 1d, and the newly added Supplementary Figure 2d, show architecture and organization of mammary ducts at one week in culture with and without endothelial cells, and we add text to the manuscript describing the non-effect of each compartment on the other for under normal conditions. We also conducted additional studies to provide a better characterization of the time course and quantification of the tissue and cellular reorganization of epithelial ducts that occurs in co-culture in Supplementary Figure 2a-c. We observe no evidence that paracrine signaling influences the establishment of our model in normal conditions, and we have added appropriate references and clarifying statements to the Results. Normal co-culture does not promote branching of either tissue compartment. We apologize for any confusion and we have updated Supplementary Figure 1c (formally 1b) with a representative example.

In the morphogen gradient studies mammary cells apparently assemble as a confluent monolayer (see Fig. 1d, vehicle) in the absence of vascular cells. Please show a staining for polarity markers on mammary duct cultured alone, where no endothelial cells are seeded. That would help to clarify the value of the co-culture. It is crucial for the authors to describe if and how the two cell compartments benefit from the co-culture in normal conditions.

The reviewer is correct that mammary cells assemble confluent, stable ducts independent of the presence of vasculature and we apologize if this was not clearly stated in the original manuscript. Similarly, the vasculature assembles independent of the epithelial compartment. In addition to the maximum intensity projections in Figure 1d and at the reviewer's suggestion, we now include a confocal slice of a stable mammary duct cultured in the absence of endothelial cells in Supplementary Figure 2d. We observe no evidence that paracrine signaling influences the establishment of our model in normal conditions and we have added appropriate references and clarifying statements to the Results.

3. Supplementary Figure 2a. The authors show an immunofluorescence staining on HMEC cells at 5 days of co-culture, however no study is mentioned in the manuscript regarding the co-culture of HMEC cells and endothelial cells. I would recommend the authors to include an assay where these two cell types are co-cultured to test if the device could be valid with a second mammary cell line.

We were in the process of conducting experiments to provide a more detailed characterization of HMEC and endothelial cell co-culture when we encountered unanticipated experimental reproducibility issues that we believe were due to phenotypic drift of our in-house primary HMECs. Ongoing and planned experiments with a new HMEC line unfortunately revealed that additional optimizations would be required to generate satisfactory results. While our preliminary data supported the use of HMECs in the system, we have removed the original Supplementary Figure 2a and reference to HMEC in the results, neither of which influence the primary conclusions and newly identified

mechanisms in this manuscript, and we plan to focus on the introduction of HMECs to our system in a future work.

4. Page 7, line 10. The authors say “...,and an invasive phenotype representative of epithelial-to-mesenchymal transition...”. To support this statement the authors should perform and show a staining for mesenchymal markers (like Vimentin, Snail) on mammary duct exposed to TGFβ1 gradient. If not it would be better to change “representative” into “suggesting”.

We performed additional analysis and quantification of the morphogenic response to FGF2 and TGFβ1 to demonstrate reproducibility and behavioral distinction and now report these in the revised manuscript (Supplementary Figure 2f). However, we were unable to complete the requested immunostains prior to the COVID-19 required shutdown of our laboratory, so we have changed the text from “representative” to “suggesting” as requested.

5. Figure 2a. It is not clear if the authors seed 100% VEGFA overexpressing cells. Please clarify.

We now include text clarification that we seed 100% GFP or VEGFA cells in the Results and Methods sections.

6. Figure 2b. Please show quantification of vascular vessel diameter with VEGFA expressing ducts in which DMSO (VEGF + Veh) or Semaxanib (VEGF + Sem) are delivered.

Quantification of vascular diameters treated with DMSO or Semaxanib in co-culture with VEGF ducts was performed and is now included in Supplementary Figure 4d.

7. Supplementary Figure 2c. Since VEGFA is a secreted protein, please show Western Blot on the supernatant (conditioned medium) of IRES-GFP and VEGF-IRES-GFP overexpressing MCF10A cells.

We agree that a Western blot of conditioned medium would be appropriate to demonstrate VEGF-A secretion. Unfortunately, we were unable to complete this planned experiment prior to the mandatory lab shutdown due to COVID. However, in support of secreted VEGF being the key mediator, we were able to complete the study suggested by this Reviewer's comment #9 demonstrating that conditioned medium from VEGF-IRES-GFP cells, but not IRES-GFP cells, stimulates vascular morphogenesis (Supplementary Figure 2c). These conditioned media experiments, along with the specific expression of VEGF in whole cell lysates (Supplementary Figure 2a), support our conclusion of vascular morphogenesis driven by secreted VEGF paracrine signaling.

8. What is the phenotype of VEGFA overexpressing cells when seeded alone and when in co-culture? The authors focus on the vascular remodelling induced by VEGFA overexpressing cells. Does VEGFA overexpression lead to any morphological/polarity state change in MCF10A cells? I would perform a staining on VEGFA overexpressing cells when seeded alone and when co-cultured.

We observed no obvious phenotypic differences in epithelial ducts due to the expression of VEGF-A or any downstream effects of VEGF-exposed vessels on the corresponding

duct co-cultures. Below are three representative phase contrast images of GFP or VEGF ducts at one week in co-culture.

Therefore, it does not appear that VEGF expression leads to obvious gain-of-function morphogenic effects on the mammary ductal epithelium.

9. To unequivocally prove that vascular remodelling is induced by VEGFA overexpressing cells via paracrine signalling I would recommend culturing vascular vessels with conditioned medium from GFP expressing ducts and VEGFA expressing ducts and perform quantification of vessel diameter and vessel sprouts in each condition.

We appreciate this suggestion. We have completed this study and now include new experimental data that demonstrates conditioned medium from VEGF-IRES-GFP cells, but not IRES-GFP cells, stimulates vascular morphogenesis (Supplementary Figure 2c). These conditioned media experiments, along with the specific expression of VEGF in whole cell lysates (Supplementary Figure 2a), support our conclusion of vascular morphogenesis driven by secreted VEGF paracrine signaling.

10. Page 10, line 9. The authors state “invasion of PI3αH1047R ducts occurred more rapidly than ErbB2^{amp} and had distinct mesenchymal morphology”. First, if the authors refer to a higher speed in the invasion process I would recommend to perform a time lapse analysis. Second, did the authors perform any staining for mesenchymal markers?

Thank you for this suggestion. We have now quantified invasion rates from multi-day time lapse analysis for each mutant duct, which confirm the higher rates of invasion by the PI3K α^{H1047R} ducts, and report the results in Supplementary Figure 5d. As suggested, we also assessed mesenchymal markers at the transcript level and demonstrate that PI3K α^{H1047R} leads to increased *SNAI1* and decreased *CDH1* mRNA. Decreased E-cadherin expression was further confirmed at the protein level by immunostain (Supplementary Figure 5g-h). Together, these results suggest that this pathway induces a more mesenchymal phenotype. These observations are now reported in the Results and Discussion sections.

11. Page 10, line 17. Please show quantification of neo-angiogenic sprouting.

Upon quantification, we observed a trend of towards increased sprouts, but the effect was not statistically significant when compared to EV or ErbB2^{amp} vessels. While we believe more extended studies will likely reveal an endothelial sprouting response to the mutant epithelial ducts, we have removed text suggesting a sprouting effect and focused this manuscript on the more dramatic effects on vessel diameter and leakiness.

12. The authors mixed mutant cells with wtype cells in a ratio 1:10. How mutant and wtype cells re-organize in the mosaic ducts? Are PI3αH1047R and Erbb2amp cells driving invasion? I would perform an anti-HA staining on PI3αH1047R and Erbb2amp ducts.

This was an excellent suggestion that led to new observations. In staining for HA, we observed that while both PI3Kα^{H1047R} and ErbB2^{amp} cells drive invasion, the *speed*, morphology, and newly identified *cellular composition* of these invasive fronts are distinct. ErbB2^{amp} invasive fronts were predominantly composed of mutant cells, yet PI3Kα^{H1047R} were composed of nearly equal ratios of mutant and wild type cells suggesting interesting nonautonomous effects of the PI3Kα^{H1047R} mutation. These behavioral effects were not dependent on the presence of vasculature. These data are now included in Supplementary Figure 5j-k and described in the Results and Discussion sections.

13. Could PI3αH1047R and Erbb2amp ducts phenotypes be dependent on signalling coming from vascular vessel? To unravel this I would suggest the authors to culture PI3αH1047R and Erbb2amp ducts alone and perform staining for actin.

This is an interesting point to consider. We now include data that show no gross morphological differences in the invasive behavior of the mutant ducts in the presence or absence of vasculature (Supplementary Figure 5i-j). However, due to COVID-19 experimental shutdown we are unable to comment on whether the rate of progression of either mutant is influenced. Nonetheless, given the consistent mutant phenotypes alone or in co-culture, our core insights into the specific morphogenic behaviors resulting from specific genetic alterations remain unchanged.

14. Figure 4a. Please show Western Blot for IL-6 and IL-6Ra on basal media (BM).

This western blot is now included in Supplementary Figure 6d and demonstrates a lack of detectable IL-6 or IL-6Rα in basal medium.

MINOR POINTS:

1. Figure 1, legend. It would be better to clearly specify in the introductory head that all the IF images shown in this figure refer to MCF10A cells at 1 week of co-culture.

This clarification has been added to the beginning of Figure 1.

2. Figure 1b. Please correct “nuceli” into “nuclei”

Thank you. This has been corrected in Figure 1.

3. Supplementary Figure 2a. I would put it in Supplementary Figure 1, since it is related to the device bioengineering and not to the morphogen gradient assay.

We appreciate the suggestion, but due to the experimental issues and COVID shutdown outlined above, we have removed the original Supplementary Figure 2a from the manuscript.

4. Figures 2a and 3b. Please describe in the section “Material and Methods” how VEGFA overexpressing MCF10A cells and PI3αH1047R/Erbb2amp MCF10A cells were maintained

before using the device and how they were seeded in the device (number of cells, culture medium). In particular, in the text (page 10, line 5) it is said mutant cells were mixed 1:10 with wild type cells. Please describe it in "Material and Methods".

Detailed information on the generation of each modified cell line, their preservation, maintenance, duration of use, and seeding have been added to the Material and Methods section.

5. Page 10, line 5. Please explain why mosaic ducts were generated. I would modify the cartoon in Figure 3a accordingly, i.e. I would use two different colours to indicate mutant and wtype cells in mosaic mammary duct.

At the reviewer's request, we have now added a different color to Figure 3a to indicate the starting mosaic of wild type and mutant cells. Additionally, we have added the following clarification as to why mosaic ducts were generated to the Results section: "Mutant cells were mixed with wild type cells at a 1:10 ratio before seeding to generate mosaic ducts, permitting the emergence and observation of aberrant mutant behaviors within the architecture of an otherwise normal duct".

Referee #3 (*Remarks to the Author and our responses*):

This study focuses on the development and characterization of a microphysiological system to investigate the crosstalk between mammary epithelial cells and their neighboring vasculature. The system consists of an endothelialized channel adjacent to a human mammary duct. Following characterization, the authors use this platform to study the effect of epithelial cell HER2/ERBB2 amplification or PIK3CA(H1047R) mutation on vascular sprouting and permeability. They suggest that IL-6 is a key player in driving endothelial cell dysfunction due to PI3K mutated mammary epithelial cells. Strengths of the manuscript include the uniqueness and simplicity of the microfluidic system and that the manuscript is clearly written. However, in its current form the model is not sufficiently well described and the biological studies are missing control conditions to substantiate the conclusions regarding the role of IL-6 in the observed vascular changes.

We appreciate this reviewer's positive critique of the study and useful suggestions. We have conducted additional experiments and improved the manuscript as a result of this critique.

Major comments:

1. The best part of the model system is that it allows studying specific aspects of the reciprocal crosstalk between epithelial cells and the vasculature under in vivo-like culture conditions while permitting isolated manipulations. Clearly, there is a tremendous need for such platforms, but the paper could do a much better job at motivating and characterizing the design of the model and demonstrate its relevance:

a. While there is some data describing transport characteristics from the vascular channel, a more complete characterization would be helpful. For example, is gradient generation stable over the culture period? How fast does the gradient establish? Also, what is the motivation for separating the vascular channel and duct 500 um apart (Krogh length is 100-200 um). Does this distance affect the gradient established between both systems?

b. Figure 2 focuses on the effect of a gradient from the duct channel on the function of the vascular channel. Therefore, it may be helpful to include transport characterization of the duct channel as well.

We agree with the reviewer that the transport characteristics were under characterized in the original submission. While gradient formation from the endothelial compartment is well characterized by diffusion, and a linear gradient was formed after one hour (Supplementary Figure 2e), the dynamics of transport from the epithelial compartment is complicated by the blunt-ended geometry. Therefore, we developed a finite element model to characterize the dynamics of mass transport from the epithelial compartment. For molecules with similar diffusion coefficients to VEGF, including IL-6, the model demonstrated that a gradient was established within two hours. While the overall gradient varied with distance from the epithelial duct, the slope of the gradient at the endothelial wall was found to be similar for the 800 μm of the endothelial tube in proximity to the tip of the epithelial duct. While the 500 μm distance between the compartments does exceed the Krogh length and is expected to influence the magnitude of the slope of the gradient, this distance was chosen to allow imaging of elaborated morphogenesis from the endothelium and epithelial ducts. We have included a new Supplementary Figure 3 with the key results from the transport model, and a description of the model and key parameters has been added to the Supplementary Methods.

c. Is there an explanation for why mutant MCF10As were seeded at a 1:10 ratio with wild type cells for experiments in Figure 3?

We have added the following clarification as to why mosaic ducts were generated to the Results section: "Mutant cells were mixed with wild type cells at a 1:10 ratio before seeding to generate mosaic ducts, permitting the emergence and observation of aberrant mutant behaviors within the architecture of an otherwise normal duct".

d. Figure 4C repeats the diffusive permeability measurements that were performed in 3C, but the magnitudes are reduced by an order of 10. Is there an explanation for this? Perhaps the difference in exposure time to conditioned media (one week co-culture vs overnight treatment with conditioned media), which should be tested with a control experiment.

We agree with this hypothesis. We were preparing an extended exposure time course to investigate this hypothesis, but our experiments were interrupted due to mandatory COVID-related laboratory shutdown. As the reviewer points out, while the conditioned media responses in Figure 4c are consistent with those observed from co-culture in Figure 3b-c across multiple metrics (tissue function, cellular morphogenic changes, molecular signatures), differences in effective IL-6 concentration or duration of IL-6 exposure in these two experimental setups may indeed dictate the extent of the endothelial behavioral response. While this interesting question would be feasible to examine in our model, it does not influence our central mechanistic conclusion that PI3K α^{H1047R} causes vascular dysfunction via increased IL-6 secretion.

e. Most of the work was performed with MCF10A, but some experiments utilized primary human mammary epithelial cells (HMECs). This comparison is valuable, but requires further explanation since HMECs appeared to form glands differently. For example, glands seemed smaller, but a direct comparison is difficult since the size of the scale bar in Suppl. Fig. S2 is missing.

We attempted to provide a more detailed description of HMEC ductal assembly, but we encountered unanticipated experimental reproducibility issues that we believe were due to phenotypic drift of our in-house primary HMECs. Ongoing and planned experiments with a new HMEC line unfortunately revealed that additional optimizations would be required to generate satisfactory results. While our preliminary data supported the use of HMECs in the system, we have removed the original Supplementary Figure 2a and reference to HMEC in the Results, neither of which influence the primary conclusions and newly identified mechanisms in this manuscript, and we plan to focus on the introduction of HMECs to our system in a future work.

2. If approached from a more biological rather than engineering design perspective, the experiments are not well-integrated and important control conditions are missing:

We appreciate the reviewer's feedback and offer important clarifications below.

a. For example, Fig. 1 shows the effect of morphogen gradients on the epithelial cells (i.e., independent of endothelial cells). Fig. 2 tests how VEGF secretion of epithelial cells affects the endothelial cells, and Figs. 3 and 4 study the effect of epithelial cells dysfunction on the vascular cells independent of VEGF. However, what is missing is how all of these aspects may be interrelated. For example, the authors show that IL-6 alters vascular permeability. These changes not only affect morphogen delivery from the vascular channel, but also suggest endothelial cell phenotypic changes that could independently affect epithelial cell invasion.

We agree with the reviewer's sentiment that we failed to seize the opportunity to pull together all of the components of the study in a more coherent systems-level view of the model. While we now do so in the revised Discussion, we also note that our experimental data suggests, at least for the particular interactions examined here, that an interactive feedback between the two compartments was not detected: In Figure 1d, the application of growth factor gradients was to assess the capacity of our mammary ducts to undergo diverse morphogenic transitions from a stable tissue architecture as a proof-of-principle, independent of paracrine crosstalk examination. In Figure 2, we observed that VEGF-expressing MCF10A ducts drove changes in the neighboring endothelium, but with new data we now confirm that expression of VEGF or the VEGF-exposed endothelium did not induce observable feedback effects on the normal MCF10A ducts (included below). In an expanded Supplementary Figure 5, we present data that suggest that mutant ducts behave with similar invasive phenotypes in the absence or presence of vasculature. Together, these results suggest that for the biological cases we examined here, the interactions appear to be one-way when present. However, as we elaborate in the revised Discussion, bidirectional effects do exist and could be recapitulated in the platform. As one example, if whole blood were present in the vasculature, we would expect that changes in vascular permeability would trigger platelet activation and inflammatory cytokine release that are known to be important potentiators of epithelial activation and cancer progression (Jain RK. *J Clin Oncol.* 2013; Labelle, Begum, Hynes. *Cancer Cell.* 2011).

b. Did treatment with Semaxanib affect the duct channel in any way? VEGF signaling has been shown to affect breast cancer behavior in an autocrine/paracrine fashion (Guo S. et al., 2010)

We performed additional experiments to address this query and observed no obvious phenotypic differences in epithelial ducts due to the expression of VEGF or downstream

of the vascular morphogenic changes in VEGF duct co-cultures. Below are three representative phase contrast images of GFP or VEGF ducts at one week in co-culture.

MCF10A are non-tumorigenic cells and thus it is unsurprising that they are unresponsive to VEGF. Therefore, it does not appear that VEGF expression leads to gain-of-function morphogenic effects on the mammary ductal epithelium.

c. A study that involves blocking IL-6 signaling in conditioned media would further solidify the claim that vascular dysfunction is driven by IL-6 secretion.

We appreciate this suggestion. In this study, we demonstrate: 1) that IL-6 secretion is specifically increased in PI3K α^{H1047R} mutant ducts, 2) PI3K α^{H1047R} mutant ducts elicit loss of vascular barrier function, 3) add-back experiments using recombinant human IL-6 cooperates in conjunction with cell-generated IL-6R α in conditioned medium to drive vascular barrier dysfunction similar to PI3K α^{H1047R} mutant ducts. This add-back experiment demonstrates the ability of IL-6 to drive vascular dysfunction, and we link increased IL-6 secretion to PI3K α^{H1047R} . While we appreciate that blocking IL-6R and downstream signaling in endothelia would provide confirmatory evidence, it is technically difficult to do so while mitigating off target effects and would not affect our central mechanistic conclusion. As a result of having limited bandwidth to complete the many requests for additional studies, especially in light of COVID-19, we were forced to prioritize other studies.

Minor comments:

1. It would be helpful to include color legends within the figures (like in some panels of Figure 1b) rather than in the figure caption.

Color legends have been added to Main and Supplementary figures where possible.

2. Figure 1d, was the effect of growth factor gradients assessed in the presence of an endothelialized vascular channel in addition to an acellular vascular channel? The presence of endothelial cells could alter gradient generation or responses to growth factors.

The application of growth factor gradients was to assess the capacity of our engineered ducts to undergo diverse morphogenic transitions from a stable tissue architecture. We

agree that the presence of an endothelialized channel would alter gradient generation and epithelial responsiveness. However, these experiments are outside the intention and scope of this report and understanding paracrine signaling from the vascular compartment to the epithelium is the focus of a separate ongoing study.

3. The use of reference 25 seems out of place in the text. The relevance of VEGF-A to pregnancy is not investigated nor referred to elsewhere in the manuscript.

Former reference 25, now reference 29, describes that VEGF is increased during pregnancy and that this increase is contributed by the mammary epithelium. In the text: “The growth factor VEGF-A (herein referred to as VEGF) is a potent angiogenic factor expressed during pregnancy²⁹ and frequently expressed during breast cancer transformation³⁰”, is being used to illustrate the biological relevance of mammary epithelium-expressed VEGF in physiologic as well as cancer contexts. We believe that this is an appropriate use of the reference.

4. Figure 2D, for clarification pls include arrows pointing to sprouts that were quantified.

The requested arrows have been added to Figure 2d.

5. For the morphological differences seen in mammary epithelial cells due to ErbB2 and PI3Kα overexpression, the quantification of cell aspect ratio, invasive area, and single/multicellular invasion in the supplementary might fit better in the main body.

We appreciate the reviewer’s suggestion. However, with the addition of new morphometric data to Supplementary Figure 5, we feel that together these data represent the morphologic differences of each mutant more effectively as a stand-alone Supplementary Figure. We believe adding these data to Figure 3 would dilute the presentation of the autonomous epithelial and nonautonomous vascular effects triggered by each mutant.

6. For supplementary Figure 4, there does not appear to be any description of sample sizes or number of tests performed.

For Supplementary Figure 4a-b (now Supplementary Figure 7a-b), the cytokine western blot dot array was performed once to identify candidate cytokines that are differentially regulated. This has been clarified in the figure legend. The critical preliminary observation from this assay, the specific increase of IL-6 with PI3Kα^{H1047R}, was validated more than three times via western blot (Figure 4a and Supplementary Figure 7d).

7. Please explain why Figs. 3 and 4 focused on actin rather than CD31 or VE-Cadherin quantification.

In our original submission, we presented micrographs and quantification of both actin (Figure 3) as well as the corresponding VE-cadherin signatures (now Supplementary Figure 6a), and actin in Figure 4. Junctional actin both influences, and is a readout of, VE-cadherin junctional stability (Polacheck & Kutys et al. Nature. 2017). Given the barrier function changes associated with each Figure, we feel that these are sufficient representations of changes in the underlying junctional cytoskeleton.

8. To demonstrate luminal filling, it would be helpful to include a cross-sectional view in addition

to the current maximum intensity projections. Also, a quantification of the data shown in Fig. 1d would be valuable to indicate reproducibility of the findings.

We now include a cross-sectional view of luminal filling for ErbB2^{amp} in Supplementary Figure 4f. In addition, we include quantification of the morphogenic behaviors in Figure 1d in Supplementary Figure 2f to further indicate distinction and reproducibility.

9. Please introduce all abbreviations at first mentioning.

We have ensured all abbreviations are introduced at first mentioning.

Reviewers' Comments:

Reviewer #1:

Remarks to the Author:

In their revised manuscript, Kutys et al., have addressed all comments from this reviewer to a satisfactory level, considering the interruption in bench-work due to the COVID-19 epidemic.

The authors have nicely integrated comments from the different reviewers to write a logical and precise report. I am particularly appreciative of the caution with which they have reported the lumen-like formation of the central hole with MCF10A cells linked to apoptosis, as "traditional acinar MCF10A model", upon request for the apoptosis test by another reviewer. Indeed, the formation of the central hole in this manner has been reported with the MCF10A cells, while in other cell models it is formed by progressive organization of cells without central apoptosis, as it is proposed to happen in vivo. In fact, the mechanisms of lumen formation as they occur in vivo are still to be deciphered.

The authors have adapted the description of the polarity status to their cell model. However, the argument that the term 'lumen' has been used by others with acinus-like structures formed by MCF10A cells (regardless of the journal in which it was published) does not make it a term that should be "commonly used" with these cells if it is not appropriate in light of the biological definition of a lumen.

Remember to use "medium" when singular (see last sentence and 5th sentence from last of new section on cell culture in the Materials and Methods)

Overall, the authors have developed a very interesting system to study genetic and microenvironmental factors that influence breast cancer onset and possibly breast cancer behavior depending on the origins of such onset. The manuscript is well-written, informative and enjoyable. We desperately need many good in vitro models to study how "normal" tissues might develop towards cancer.

Sophie Lelievre

Reviewer #2:

Remarks to the Author:

The revision made by Kutys and colleagues has addressed most of the concerns I raised in the original review. The new data shown allow the readers to appreciate the value of the 3D device and understand in details the morphological modifications occurring in both cell compartments.

Supp. Fig. 8 should be amended. No Supp. Fig. 6c and Supp. Fig. 6d are shown in the revised manuscript. These two uncropped western blots should refer to Supp. Fig. 7c and Supp. Fig. 7d.

To conclude, I think the manuscript is now suitable for publication.

Reviewer #3:

Remarks to the Author:

The authors have appropriately addressed most questions from the previous round of reviews. These changes have clarified the paper and increased its overall impact. It is unfortunate that experiments with the primary cells were not repeatable and were thus, removed from the revised version. These data could have validated broader applicability of the model and results. However,

heterogeneity within primary cells is to be expected, and together with the COVID-19 pause the author's explanation may suffice as a justification.

Response to Remaining Reviewer Comments

Referee #1 (*Remarks to the Author* and our responses):

In their revised manuscript, Kutys et al., have addressed all comments from this reviewer to a satisfactory level, considering the interruption in bench-work due to the COVID-19 epidemic.

The authors have nicely integrated comments from the different reviewers to write a logical and precise report. I am particularly appreciative of the caution with which they have reported the lumen-like formation of the central hole with MCF10A cells linked to apoptosis, as “traditional acinar MCF10A model”, upon request for the apoptosis test by another reviewer. Indeed, the formation of the central hole in this manner has been reported with the MCF10A cells, while in other cell models it is formed by progressive organization of cells without central apoptosis, as it is proposed to happen in vivo. In fact, the mechanisms of lumen formation as they occur in vivo are still to be deciphered.

The authors have adapted the description of the polarity status to their cell model. However, the argument that the term ‘lumen’ has been used by others with acinus-like structures formed by MCF10A cells (regardless of the journal in which it was published) does not make it a term that should be “commonly used” with these cells if it is not appropriate in light of the biological definition of a lumen.

Remember to use “medium” when singular (see last sentence and 5th sentence from last of new section on cell culture in the Materials and Methods)

Overall, the authors have developed a very interesting system to study genetic and microenvironmental factors that influence breast cancer onset and possibly breast cancer behavior depending on the origins of such onset. The manuscript is well-written, informative and enjoyable. We desperately need many good in vitro models to study how “normal” tissues might develop towards cancer.

Sophie Lelievre

We were pleased to receive such positive assessment of our work. Thank you for your consideration and insightful suggestions, which have improved the quality of this manuscript. We have corrected the two instances to use the term “medium” in the Materials and Methods. In addition, we have modified the description of our device in the Results considering your suggestions regarding use of the word lumen:

“The resulting stable, noninvasive monolayers within the channels displayed growth arrest as evidenced by minimal actively proliferating Ki67+ cells (Fig. 1b), a process that is dependent upon stable cell-cell contacts in MCF10A²⁴⁻²⁶. Thus, despite the limited maturation capabilities of MCF10As, this approach resulted in an epithelial cell-lined dead-ended channel, reminiscent of an anatomical ductal lumen.”

Referee #2 (*Remarks to the Author* and our responses):

The revision made by Kutys and colleagues has addressed most of the concerns I raised in the original review. The new data shown allow the readers to appreciate the value of the 3D device and understand in details the morphological modifications occurring in both cell compartments.

Supp. Fig. 8 should be amended. No Supp. Fig. 6c and Supp. Fig. 6d are shown in the revised manuscript. These two uncropped western blots should refer to Supp. Fig. 7c and Supp. Fig. 7d.

To conclude, I think the manuscript is now suitable for publication.

We have fixed the incorrect labeling in Supplementary Figure 8. We thank the reviewer for their considerations and helpful comments.

Referee #3 (*Remarks to the Author* and our responses):

The authors have appropriately addressed most questions from the previous round of reviews. These changes have clarified the paper and increased its overall impact. It is unfortunate that experiments with the primary cells were not repeatable and were thus, removed from the revised version. These data could have validated broader applicability of the model and results. However, heterogeneity within primary cells is to be expected, and together with the COVID-19 pause the author's explanation may suffice as a justification.

We thank the reviewer for their consideration and helpful comments that improved the quality of this work.